# Probabilistic Hash Embeddings for Temporal Tabular Data Streams

## Abstract

We study *temporal tabular data-streams* (TTD) where each observation has both categorical and numerical values, and where the universe of distinct categorical items is not known upfront and can even grow unboundedly over time. Such data is common in many large-scale systems, such as user activity in computer system logs and scientific experiment records. Feature hashing is commonly used as a pre-processing step to map the categorical items into a known universe, before doing representation learning (Coleman et al., 2024; Desai et al., 2022). However, these methods have been developed and evaluated for the offline or batch settings. In this paper, we consider the pre-processing step of hashing before representation learning in the online setting for TTD. We show that deterministic embeddings suffer from forgetting in online learning with TTD, leading to performance deterioration. To mitigate the issue, we propose a *probabilistic hash embedding* (PHE) model that treats hash embeddings as stochastic and applies Bayesian online learning to learn incrementally with data. Based on the structure of PHE, we derive a scalable inference algorithm to learn model parameters and infer/update the posteriors of hash embeddings and other latent variables. Our algorithm (i) can handle evolving vocabulary of categorical items, (ii) is adaptive to new items without forgetting old items, (iii) is implementable with a bounded set of parameters that does not grow with the number of distinct observed items on the stream, and (iv) is efficiently implementable both in the offline and the online streaming setting. Experiments in classification, sequence modeling, and recommendation systems with TTD demonstrate the superior performance of PHE compared to baselines.

## 1 Introduction

Tabular data - where each observation is a vector with both categorical and numerical values - is very common. For example, tabular data can be any records in a MS Excel file, songs information in a music playlist, execution results of the Linux command "ls -l," and any tables seen in this paper. As a result, tabular data occurs in many high-valued ML applications: finance (Clements et al., 2020), fraud detection (Al-Hashedi and Magalingam, 2021), anomaly detection (Han et al., 2022), cybersecurity (Sarker et al., 2020), medical diagnosis (Shehab et al., 2022) and recommendation systems (Ko et al., 2022). In many of these applications, the data arrives online in a streaming fashion.

Unlike images and natural language text, tables are highly structured and contain heterogeneous data types that often result in a mix of both categorical and numeric features (Borisov et al., 2022; Shwartz-Ziv and Armon, 2022). Recent work on tabular data focuses on designing generative models for tabular data type (Xu et al., 2019; Kotelnikov et al., 2023; Liu et al., 2023b) or learning table representations with foundation models (Yin et al., 2020; Iida et al., 2021). However, these are *offline* models that assume that the characteristics such as the vocabulary of columns are fixed.

In *temporal* tabular data (TTD), *(i)* the vocabulary of some or all categorical columns can change, and/or *(ii)* the semantic meaning of a categorical item can evolve. These characteristics present challenges to offline predictive models. Failure to adapt to the expanding vocabulary leads to a loss in predictive performance, as explained in Figure 1. In the setup of Figure 1, we observe that modeling new categorical features leads to significant accuracy improvement. This problem of expanding vocabulary is fairly common in practice: new products are added to a grocery store (Cheng et al.,

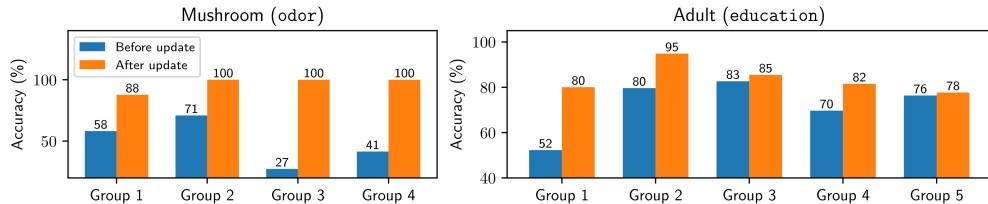

Figure 1: On two tabular datasets, Mushroom and Adult, we split the data into groups based on a random partition of a categorical column's vocabulary, such that each group has a disjoint vocabulary. We report the results before and after online learning on each group in the plots. The performance gaps motivate the need to learn representations of new items. In the brackets are the columns used for splitting. Results are averaged on five independent runs. The partition detail is in appendix Fig. 9.

2023), new usernames and application names in intrusion detection systems (Siadati and Memon, 2017; Le et al., 2022), new patients at a hospital, and so on.

The expanding and unbounded universe of categorical values[1] poses challenges, even in the offline setting when all training data is available upfront and the size of the vocabulary can be billions (Tito Svenstrup et al., 2017; Shi et al., 2020b; Kang et al., 2021; Coleman et al., 2024). A commonly used methodology to handle unbounded vocabulary is the *hashing trick* (Weinberger et al., 2009), where one or more hash functions map the categorical values to a value in a fixed finite set. The hashed values are treated as an approximation of the original categorical values in subsequent model training and inference. The resulting item representations are stored in a set of model parameters, referred to as *hash embeddings*. Large technology firms, e.g., Yahoo and Google, have incorporated this approach in their large-scale applications (Weinberger et al., 2009; Coleman et al., 2024).

While hash embeddings claim to handle "dynamic" vocabularies, previous work focuses on offline settings. In this paper, we study learning hash-based embeddings in TTD, whose categorical vocabulary is really *dynamic*, i.e., changing over time. We analyze and demonstrate hash embeddings are subject to catastrophic forgetting as the vocabulary grows. In hash embeddings, representations of two items may share parameters, updating one item's embedding can adversely interfere with another, causing an effect like the model "forgets." Consequently, hash embeddings are not yet fit for learning temporally dynamic vocabularies in its vanilla form.

In this paper, we observe that modeling hash embeddings as stochastic and inferring their posterior upon new data arrival mitigates the shortcomings found in online update of deterministic hash embeddings. This Bayesian online learning approach is shown as sample-efficient as offline batch learning (Opper and Winther, 1999), which can in turn make the hash embeddings estimate as effective as offline training.

**Main Contributions:** Our work proposes *probabilistic hash embeddings* (PHE) with Bayesian updates to handle dynamic vocabularies of TTD in an effective and efficient way. The intuition behind PHE stems from its benefits in *(i)* efficiency, as memory/number of model parameters is bounded and only a small number of parameters need to be updated online (ie., less forgetting, see experiments), and *(ii)* accuracy benefits since the probabilistic model provides an implicit regularization to trade-off forgetting and adaptation, without the need of specific dataset dependent regularization design.

We highlight PHE as a plug-in module, which can be applied to other probabilistic models like Deep Kalman Filters (Krishnan et al., 2015), a latent variable model for temporal sequences, and Neural Collaborative Filtering (He et al., 2017), modeling item-user interactions in recommendation systems. The usage of PHE allows those models to handle unbounded items in their application areas in a principal way. For those models, we derive scalable variational inference algorithms to learn the model parameters and infer the latent variables (including latent time variables and PHE). Empirically, we observe superiority of our method compared to baselines under three setups: one supervised learning where new items occur in sequence, the second is conditional sequence modeling setup where the number of sequences increases along with new items, and the third is a recommendation system where novel user-item interactions occur over time.

---

[1]We use the phrase *item* to refer to a categorical value.

**Organization:** We survey related work in Sec. 2, present PHE, derive its inference algorithm in Sec. 3 and demonstrate PHE's efficacy in Sec. 4 and conclude in Sec. 5.

## 2 RELATED WORK

**Hashing trick.** Weinberger et al. (2009) first proposed using hashing to handle unbounded number of categorical items. To improve on degradation due to hash collisions, Serrà and Karatzoglou (2017) used bloom filters. In recent times, Tito Svenstrup et al. (2017); Cheng et al. (2023); Coleman et al. (2024) propose a shared embeddings across all categorical features for efficiency and using multiple hashing functions to reduce collisions. However, unlike our method, these are deterministic and are developed in offline learning settings. As shown in our experiments, deterministic hashing embeddings are vulnerable to evolving vocabularies and inter-observation relation shifts in TTD.

**Continual learning.** Wang et al. (2023) surveys main works in continual learning. Regularization-based methods (EWC (Kirkpatrick et al., 2017), VCL Nguyen et al. (2018)) and optimization-based methods (e.g., GEM (Lopez-Paz and Ranzato, 2017)) ignore categorical variables with unbounded vocabulary size. Architecture-expanding methods that dynamically expand the universe of items leads to unbounded memory usage (Rusu et al., 2016; Yoon et al., 2017; Jerfel et al., 2019). In contrast, PHE uses a fixed size of memory to accommodate expanding categorical features.

**Temporal and recommendation models.** One of our models extends Deep Kalman Filters (DKF) (Krishnan et al., 2015) to be applicable for sequence modeling in TTD, while the original DKF only apply to time-series data where the categorical attributes are assumed to be given and thus not modeled. Girin et al. (2021) survey a list of latent variable sequence models for speech, text and video and are not applicable to tabular data due to the ever increasing vocabulary of categorical items. Similarly previous recommendation methods (Ko et al., 2022) assume the training data is given at once and the universe of items is stationary.

**Tabular data models.** Traditionally, tabular data refers to rows in a database, whose distribution are permutation invariant (Friedman, 2001). In the offline setting when the universe of categorical values are known, tree-based boosting methods have emerged as competitive (Chen and Guestrin, 2016; Ke et al., 2017). In the online and continual learning setting, deep-learning based methods have been studied in recent times (Huang et al., 2020; Du et al., 2021; Liu et al., 2023a). However, all of these works assume that the universe of categorical values are known and fixed up-front. *Ours is the first online learning method, even for regular tabular data, that can handle increasing and unbounded vocabulary for items.* Kim et al. use string embeddings from language models for open-vocabulary categorical/string-valued columns in an offline setting; in contrast, we focus on online setting. We survey additional related work in Supp. B.

## 3 METHODOLOGY

We first set necessary notations, then introduce our proposed probabilistic hash embedding module. Next we show that PHE as a plug-in module for Deep Kalman Filters. Finally we analyze why deterministic hash embeddings is prone to forgetting.

### 3.1 PROBLEM SETUP AND NOTATIONS

We formalize the notations here. We denote categorical, numeric, and timestamp columns or their feature values by $\mathbf{s}, \mathbf{x}, t$ respectively. For the columns of interest that we want to predict based on other columns, we denote them by $\mathbf{y}$. We use $\mathbf{s}_i$ to denote the categorical values of the $i$th row, similarly for $\mathbf{x}_i, t_i$, and $\mathbf{y}_i$. We consider the problem of learning an ML model in TTD streams where the vocabulary of one or more categorical columns can change over time.[2]

Let $h : \mathcal{S} \to \mathbb{N}_{<B}$ be a hash function that maps a string to a hash value. $B$ is the range of $h$, also known as the "bucket size". For simplicity, we use $h_\mathbf{s}$ to denote the hash value $h(\mathbf{s})$ of an item $\mathbf{s}$. The hash value $h_\mathbf{s}$ indexes a row in a hash embedding table $E \in \mathbb{R}^{B \times d}$, resulting in the hash embedding

---

[2]We assume the tabular structure is fixed, i.e., the number of columns, column names, and types are fixed. We also assume categorical features are single-valued. But our work is compatible with multi-valued features.

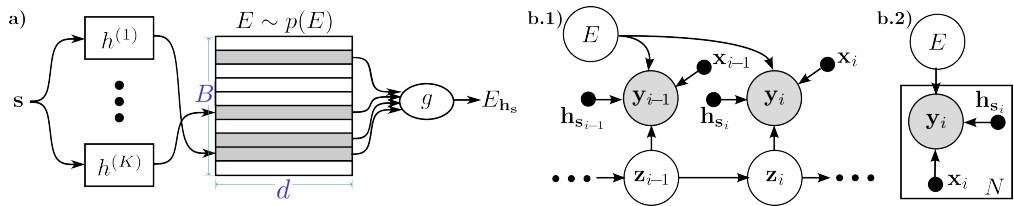

Figure 2: **a)** Shared hash embedding for category feature **s**. For example, **s** can be a username or anonymized string. The whole module serves as $p(E_{\mathbf{h_s}})$. **b.1)** A graphical model of a temporal sequence with PHE. **b.2)** is a special case of **b.1** when no temporal relationship is assumed, e.g., classification or regression. The changing categorical values are contained in $\mathbf{h_s}$.

of **s**, denoted by $E_{h_{\mathbf{s}}}$. We abuse the notation for both random variables and their sampled values where the meaning should be clear from the context.

## 3.2 PROBABILISTIC HASH EMBEDDINGS (PHE)

To start off, we explain our universal encoding module for categorical items. Categorical items are the most common ones in tabular datasets, ranging from login names, database names, and activity codes to anonymized identifiers, and are subject to increment from time to time. To incorporate the added items, we propose probabilistic hash embeddings (PHE). The basic PHE involves two components–a fixed hash function $h \in \mathbb{N}_{<B}$ and a probabilistic hash embedding table $E \in \mathbb{R}^{B \times d}$ with a prior distribution $p(E)$.[3] Given an item **s**, it looks up the $h_{\mathbf{s}}$th row of $E$ as its hash embedding $E_{h_{\mathbf{s}}}$. $E_{h_{\mathbf{s}}}$ has distribution $p(E_{h_{\mathbf{s}}})$. We require $E$ to be shared across all categorical columns, an operation adopted by Coleman et al. (2024) as well. To disambiguate duplicated feature values across columns, we add the column name as a prefix to its items. Thus, the same string in different columns will likely be hashed to different values, reducing hash collisions. PHE can also learn the representations of missing values, which are usually represented as a special string and can be hashed.

A single hash function may result in two distinct inputs having the same hashing value, known as hash collisions, resulting in undistinguished hash embeddings. For size-$B$ buckets, the collision probability is proportional to $O(1/B)$. To further reduce the collision rate, we use universal hashing (Carter and Wegman, 1977). Namely, instead of utilizing one fixed hash function, we use $K$ fixed hash functions. Then the collision probability can be shown to reduce to $O(1/B^K)$. Moreover, we keep the hash embedding table $E$ shared across $K$ hash functions, which keeps the model size bounded. Repeating the embedding fetching procedure, a single feature **s** results in $K$ embeddings $\{E_{h_{\mathbf{s}}^{(1)}}, \ldots, E_{h_{\mathbf{s}}^{(K)}}\}$ where $E_{h_{\mathbf{s}}^{(k)}}$ is the looked-up embedding from table $E$ based on the $k$-th hash value $h_{\mathbf{s}}^{(k)}$. We then produce the final representation of **s** with an assemble function $g : \mathbb{R}^{K \times d} \to \mathbb{R}^d$. This procedure is denoted by $E_{\mathbf{h_s}} := g(E_{h_{\mathbf{s}}^{(1)}}, \ldots, E_{h_{\mathbf{s}}^{(K)}})$. Typical choices of $g$ involve coordinate-wise summation, average, and minimization; other parametric choices of $g$ include weighted sums where weights come from a parametric model. We illustrate this procedure in Fig. 2a. This module represents $p(E_{\mathbf{h_s}})$ where $\mathbf{h_s} := \{h_{\mathbf{s}}^{(1)}, \ldots, h_{\mathbf{s}}^{(K)}\}$. The memory cost of PHE is $O(Bd)$.

A common query is to ask what the conditional probability of observing a feature **y** given another categorical feature **s** is, namely $p(\mathbf{y}|\mathbf{s})$. With PHE, we can approximate it by identifying $\mathbf{h_s}$ to be **s**, which is exact in the absence of hash collisions. Thus with $E_{\mathbf{h_s}}$ sampled from $p(E_{\mathbf{h_s}})$,

$$p(\mathbf{y}|\mathbf{s}) = p(\mathbf{y}|\mathbf{h_s}) = \mathbb{E}_{p(E)}[p(\mathbf{y}|E_{\mathbf{h_s}})] \approx p(\mathbf{y}|E_{\mathbf{h_s}}). \tag{1}$$

In this way, one can answer probability queries conditioned on discrete features.

**Discussions.** In data streaming or continual learning setup, PHE has natural benefits in reducing catastrophic forgetting: 1) only a few embeddings need to be updated online. This sparse updating scheme seldom affects other item representations, thus having less forgetting. 2) The online updates apply Bayesian online learning, in which the prior distribution serves as a regularization of previous knowledge that also reduces forgetting. In addition, PHE's memory/storage cost is bounded and does not increase with the number of distinct categorical values. More discussion in Sections 3.4 and D.

---

[3]In this work, we assume $E$ is Gaussian with a diagonal covariance.

## 3.3 An Application: PHE in Deep Kalman Filters

In this section, we show how PHE can be used in conjunction with a deep Kalman filter (Krishnan et al., 2015) and derive scalable inference algorithms for the probabilistic embedding. Other model variants for TTD can be seen as special cases of this model. The goal is to predict $\mathbf{y}$, given all other columns including categorical columns $\mathbf{s}$, numeric columns $\mathbf{x}$, and timestamp column $t$. (c.f. Fig. 2b.1.) The condition of $\mathbf{y}$ on categorical features $\mathbf{s}$ is through PHE via Eq. (1). We model the dependency between rows of neighboring timestamps by a latent time variable $\mathbf{z}$. Specifically, we assume $\mathbf{z}_i$ are Gaussian distributed and follows the distribution $p(\mathbf{z}_i|\mathbf{z}_{i-1}, \Delta_i; \theta_z) = \mathcal{N}(\mathbf{z}_i|f_{\theta_z}(\mathbf{z}_{i-1}, \Delta_i))$ where $f_{\theta_z} := \{\mu_{\theta_z}, \Sigma_{\theta_z}\}$ is a parametric function with parameters $\theta_z$, e.g., a multi-layer perceptron that outputs mean and covariance of $\mathbf{z}_i$ and $\Delta_i$ is the difference in timestamp between the $i$th observation and $i-1$th observation. We apply a diagonal covariance matrix $\Sigma_{\theta_z}$ in this work. We assume the initial row's latent representation $\mathbf{z}_1$ are from standard Gaussian distribution $p(\mathbf{z}_1) = \mathcal{N}(\mathbf{z}_1|0, I)$. This usage of a latent time variable shares a similar fashion with Kalman filters.

In summary, suppose a parametric likelihood with parameters $\theta_y$ is $p(\mathbf{y}_i|\mathbf{x}_i, E_{\mathbf{h}_{\mathbf{s}_i}}; \theta_y)$. Given the covariates $\mathbf{h}_{\mathbf{s}_{\leq N}}, \mathbf{x}_{\leq N}$, and time difference $\Delta_{\leq N}$, the data generating process is

$$E \sim p(E), \quad \text{For } i = 1, \ldots, N: \quad \left\{\mathbf{z}_i \sim p(\mathbf{z}_i|\mathbf{z}_{i-1}, \Delta_i; \theta_z), \ \mathbf{y}_i \sim p(\mathbf{y}_i|\mathbf{x}_i, E_{\mathbf{h}_{\mathbf{s}_i}}; \theta_y)\right\}$$

where $p(\mathbf{z}_1) = \mathcal{N}(0, I)$. Observations of other tasks are generated similarly beside the hash embedding table $E$ and the parameters $\{\theta_z, \theta_y\}$, which are shared across tasks.

**Inference network.** In the above model, we need to infer the hash embedding table and latent time variables, that is, the posterior distribution $p(E, \mathbf{z}_i|\mathbf{h}_{\mathbf{s}_{\leq i}}, \mathbf{x}_{\leq i}, \mathbf{y}_{\leq i}; \theta)$ after observing $i$ rows, which is often intractable with complex likelihood and expensive for large-scale datasets. Therefore, we apply structured variational inference and assume the variational posterior distribution factorizes as

$$q_{\lambda, \phi}(E, \mathbf{z}_{\leq N}|\mathbf{x}_{\leq N}, \mathbf{y}_{\leq N}, \mathbf{h}_{\mathbf{s}_{\leq N}}) = q_\lambda(E) \prod_{i=1}^{N} q_\phi(\mathbf{z}_i|\mathbf{x}_{\leq i}, \mathbf{y}_{\leq i}, E_{\mathbf{h}_{\mathbf{s}_{\leq i}}}) \tag{2}$$

where we parameterize the posterior of the hash embedding table as a Gaussian with diagonal covariance, i.e., $q_\lambda = \mathcal{N}(\mu_\lambda, \Sigma_\lambda)$ with variational parameters $\lambda := \{\mu_\lambda \in \mathbb{R}^{B \times d}, \Sigma_\lambda \in \mathbb{R}^{B \times d}\}$. We also assume $q_\phi(\mathbf{z}_i|\mathbf{x}_{\leq i}, E_{\mathbf{h}_{\mathbf{s}_{\leq i}}})$ is a Gaussian distribution implemented as a recurrent neural network that takes $\{\mathbf{x}_i, \mathbf{y}_i, E_{\mathbf{h}_{\mathbf{s}_i}}\}$ as input at recurrent step $i$ and outputs the parameters of $\mathbf{z}_i$, in this case, mean $\mu_{i,\phi} \in \mathbb{R}^d$ and diagonal covariance matrix $\Sigma_{i,\phi} \in \mathbb{R}^d$. Note that the recurrent neural network (i.e., its parameters $\phi$) is shared across latent time variables $\mathbf{z}_{\leq N}$.

**Initialization and online learning.** As follows, we will first derive a scalable algorithm to initialize the model with a set of training data and then introduce an efficient online learning algorithm to adapt the model to TTD. We denote the model parameters relevant to the generating process by $\theta := \{\theta_z, \theta_y\}$. We jointly learn the model parameters $\theta$ and infer the approximate posteriors of $\{E, \mathbf{z}_{\leq N}\}$ by maximizing a feasible evidence lower bound (ELBO) $\mathcal{L}(\theta, \lambda, \phi)$. It can be shown that, with twice the applications of Jensen's inequality, maximizing the ELBO also maximizes the marginal likelihood $\log p(\mathbf{y}_{\leq N}|\mathbf{x}_{\leq N}, \mathbf{h}_{\mathbf{s}_{\leq N}}; \theta)$

$$\mathcal{L}(\theta, \lambda, \phi) := \mathbb{E}_{q_\lambda(E)} \left[\sum_{i=1}^{N} \mathcal{L}_i(\theta, \phi|E)\right] - D_{\mathrm{KL}}(q_\lambda(E)|p(E)) \leq \log p(\mathbf{y}_{\leq N}|\mathbf{x}_{\leq N}, \mathbf{h}_{\mathbf{s}_{\leq N}}; \theta) \tag{3}$$

where $p(E)$ is the prior distribution of the random hash embedding table and we set it to be a standard Gaussian distribution. $\mathcal{L}_i(\theta, \phi|E)$ is the conditional ELBO of the $i$th row's log-likelihood

$$\mathcal{L}_i(\theta, \phi|E) := \mathbb{E}_{q_\phi(\mathbf{z}_i)}[\log p(\mathbf{y}_i|\mathbf{z}_i, \mathbf{x}_i, E_{\mathbf{h}_{\mathbf{s}_i}}; \theta_y)] - \mathbb{E}_{q_\phi(\mathbf{z}_{i-1})}[D_{\mathrm{KL}}(q_\phi(\mathbf{z}_i)|p(\mathbf{z}_i|\mathbf{z}_{i-1}; \theta_z))]. \tag{4}$$

We provide the full derivation of the ELBO in Supp. A, where we also show $\mathcal{L}_i(\theta, \phi|E) \leq \log p(\mathbf{y}_i|\mathbf{x}_{\leq i}, \mathbf{y}_{<i}, E_{\mathbf{h}_{\mathbf{s}_{\leq i}}}; \theta)$ and why maximizing $\mathcal{L}(\theta, \lambda, \phi)$ is a variational EM algorithm.

Once we initialize the model, we can efficiently adapt the model to new occurring items in TTD. We only need to update the hash embeddings of new items while fixing other parts of the model. This scheme is feasible since per the proposed data generating process, the latent time process $\mathbf{z}_i$ and model parameters $\theta$ has captured all item-shared information and the item-specific information is to be learned via the new hash embedding.

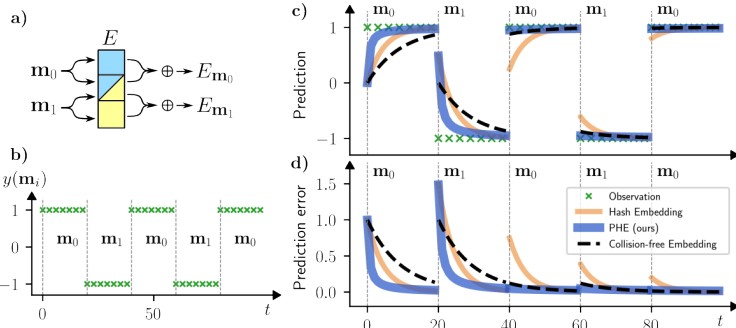

Figure 3: Forgetting in online learning using deterministic hash embedding on synthetic data. (The complete setting is described in Sec. 3.4.) The task is predicting a scalar (regression problem) with the covariate being a categorical variable that takes one of two values of $\mathbf{m}_0$ or $\mathbf{m}_1$. **a)** shows the embedding matrix $E$ of size $3 \times 1$. Here the number of buckets $B = 3$ and $d = 1$. The two hash function maps $\mathbf{m}_0$ to 0 and 1 respectively and maps $\mathbf{m}_1$ to 1 and 2 respectively. **b)** shows the online samples where the covariate alternates between $\mathbf{m}_0$ and $\mathbf{m}_1$ and the corresponding target $y(\mathbf{m}_i)$ takes values in 1 and $-1$. **c)** shows the prediction of a probabilistic hash embedding table (blue) trained using Bayesian online learning and a deterministic hash embedding (DHE) table (yellow) trained using online gradient descent. **d)** plots the prediction error. From these figures we observe that PHE's prediction error converges to 0 much quicker than DHE. After every 20 samples when the covariate changes, there is a big jump in DHE error, exhibiting forgetting while the PHE has no error spikes after it has encountered both the categorical values.

Suppose the initialization dataset $\mathcal{D}_0$ is large enough to allow to infer good model parameters $\{\lambda_0^*, \phi^*, \theta^*\}$. (The subscript of $\lambda_0^*$ suggests the posterior of hash embedding table $E$ is conditional on dataset $\mathcal{D}_0$.) Suppose we observe a second dataset $\mathcal{D}_1$ to which we would like our model to adapt and still be effective to $\mathcal{D}_0$. We can set the posterior distribution $q_{\lambda_0^*}(E)$ of hash embeddings as the new prior distribution and infer the new posterior of $E$ conditioned on $\mathcal{D}_1$. Note that $\theta^*$ and $\phi^*$ are fixed during this adaptation procedure. The new ELBO (objective function) on $\mathcal{D}_1$ given $\theta^*, \lambda_0^*, \phi^*$ is

$$\mathcal{L}^{(1)}(\lambda; \theta^*, \lambda_0^*, \phi^*) = \mathbb{E}_{q_\lambda(E)} \left[ \sum_{i=1}^{N_1} \mathcal{L}_i(\theta^*, \phi^*|E) \right] - D_{\mathrm{KL}}(q_\lambda(E)|q_{\lambda_0^*}(E)) \tag{5}$$

where $N_1$ is the number of rows in table $\mathcal{D}_1$. Notice the original prior $p(E)$ of $E$ is replaced with $q_{\lambda_0^*}(E)$. Upon optimization convergence, the new variational distribution of $E$ is equivalent to an approximate posterior distribution given both datasets ($\mathcal{D}_0$ and $\mathcal{D}_1$). We provide detailed derivations in Supp. A. Although this procedure bears resemblance to traditional continual learning (Wang et al., 2023; Nguyen et al., 2018; Li et al., 2021), is different since we focus on changing discrete items.

## 3.4 THEORY: WHY IS PHE SUPERIOR FOR TTD?

Here, we consider a simple linear Gaussian model that we can analyze in closed form to illustrate why having deterministic hash embeddings that are updated in an online fashion is prone to *forgetting*.

**A simple linear-Gaussian model.** Let the input variable $\mathbf{m} \in \{\mathbf{m}_0, \mathbf{m}_1\}$ take one of two categorical values and the target $y \in \mathbb{R}$ be real-valued. The conditional distribution of $y$ is a Gaussian distribution with the mean 1 when $\mathbf{m} = \mathbf{m}_0$ and mean $-1$ when $\mathbf{m} = \mathbf{m}_1$. The variance of both Gaussians is $\sigma^2 \approx 0$ is tiny. We do not specify the distribution of $\mathbf{m}$ just yet and defer that to the sequel.

**The predictive model based on hash embedding.** Given labeled data $(\mathbf{m}, y)$, we aim to learn a predictor $f(\mathbf{m})$ for $y$ using a $3 \times 1$ hash embedding matrix[4] $E$. Denote by $e^{(0)}, e^{(1)}$ and $e^{(2)}$ as the three rows of this matrix which are the "embedding vectors" of the three hash values. Thus, in the notation of our model, this embedding matrix is made of $B = 3$ buckets with the dimension $d = 1$. The model $f(\cdot)$ uses two hash functions $h_i(\cdot) : \{\mathbf{m}_0, \mathbf{m}_1\} \to \{0, 1, 2\}$ to map the categorical variable $\mathbf{m}$ into a hash value. Without loss of generality, we assume that $h_1(\mathbf{m}_0) = 0, h_2(\mathbf{m}_0) =$

---

[4]Although technically a vector, we denote it as "embedding matrix" to be consistent with the rest of the text.

$1, h_1(\mathbf{m}_1) = 1, h_2(\mathbf{m}_1) = 2$. Given this, the predictive model $f(\mathbf{m}) := e^{(h_1(\mathbf{m}))} + e^{(h_2(\mathbf{m}))}$ is a linear sum of the two hash embedding of the input. This is a simple example of the general class of models where the predictor $y$ is a linear function of the embedding vectors looked up by the categorical input $\mathbf{m}$ using different hash functions. Although simple, this example illustrates the *parameter-interference* phenomenon of hash embeddings since the embedding vector $e^{(1)}$ influences both $\mathbf{m}_0$ through hash function $h_1(\cdot)$ and $\mathbf{m}_1$ through $h_2(\cdot)$.

**An online interaction setting.** At each time $t = 1, 2, \ldots$, the environment samples $\mathbf{m}^{(t)}$ from a distribution over $\{\mathbf{m}_0, \mathbf{m}_1\}$ and sends to the predictor. The predictor then predicts $\widehat{y}_t := f_t(\mathbf{m}^{(t)})$ and is then shown the true label $y \in \mathbb{R}$. The predictor incurs loss $l_t := \frac{1}{2}(y_t - \widehat{y}_t)^2$ and uses the observed $y_t$ to update the predictor to $f_{t+1}(\cdot)$. Consider a setting where the first $N$ inputs consists of $\mathbf{m}^{(t)} = \mathbf{m}_0$ for all $t \in \{1, \ldots, N\}$, followed by another $N$ inputs consisting of $\mathbf{m}^{(t)} = \mathbf{m}_1$ for all $t \in \{N + 1, \cdots, 2N\}$. The embedding matrix $E$ is assumed to be updated using online gradient descent on the square loss function $l_t := \frac{1}{2}(\widehat{y}_t - y_t)^2$ for all times $t = 1, 2, \ldots$.

**Analysis and conclusion.** As we can see from the calculations in Supp. D, if $N$ is sufficiently large, at the end of time $2N$, the learned model is such that $f_{2N+1}(\mathbf{m}_0) \approx 1/4$ and $f_{2N+1}(\mathbf{m}_1) \approx -1$. Thus, given that the last $N$ samples seen corresponded to $\mathbf{m}^{(t)} = \mathbf{m}_1$, the predictor at the end at time $2N$ has near zero prediction error for $\mathbf{m}_1$. However, this comes at a cost of having a large prediction error for $\mathbf{m}_0$ with $f_{2N+1}(\mathbf{m}_0) \approx 1/4$, where the true value is 1. This is in contrast to an offline method that given all the $2N$ samples upfront, the algorithm would have learned a predictor that will have near zero prediction error for both $\mathbf{m}_0$ and $\mathbf{m}_1$. Thus, we say that the online updated embedding matrix *forgets* the old distribution $\mathbf{m}_0$. This behaviour is also in contrast to Bayesian online learning that would incrementally learn the posterior distribution $p(E|\mathbf{m}^{(1)}, \ldots, \mathbf{m}^{(t)})$ at each time $t$. It is well known that if we could compute the exact posterior at each time, i.e., exactly compute $p(E|\mathbf{m}^{(1)}, \ldots, \mathbf{m}^{(t)})$ for all $t$, then the posterior for $E$ at the end of $2N$ samples will be identical to the case if all the $2N$ samples would be available up-front in batch, i.e., there will be no forgetting. Fig. 3 shows an example when $N = 20$. Detailed calculations are in Supp. D.

## 4 EXPERIMENTS

In this section, we show TTD is ubiquitous in mainstream machine learning tasks and conduct experiments to demonstrate the efficacy and memory efficiency of PHE in learning TTD. As follows, we begin with common experimental protocols in Sec. 4.1. Then, in Sec. 4.2, we simulate online learning to benchmark PHE in classification tasks. In Secs. 4.3 and 4.4, we showcase PHE in multi-task sequence modeling and online recommendation systems. All results show PHE outperforms the deterministic counterpart and performs similarly with the upper-bound collision-free embeddings.

### 4.1 EXPERIMENTAL PROTOCOLS AND BASELINES

**Training protocols.** In all experiments, we use one shared hash embedding table, also known as unified embeddings (Coleman et al., 2024), for all categorical features. Except for the initial training, we only update the embeddings of categorical columns deemed incremental and freeze parameters other than the hash embedding table. Tab. 6 in Supp. E.6 justifies this updating protocol.

**Experimental setting.** We investigate the data-streaming setup highlighting recurring items and new-arriving items. In this setup, both forgetting and adaptation are measured: the model should avoid forgetting recurring items and adapt for new items. Upon each data arrival, we conducted three operations in order: make predictions, evaluate predictions, and update embeddings. We report the sequential results in plots and the overall averaged results with errors in tables. We repeated all experiments five times with different parameter initialization while keeping other settings fixed.

**Baselines.** We use two types of baselines. The first is deterministic hash embeddings with stochastic gradient descent online learning. They not only have the same model size and architecture as PHE but also have the same update efficiency. That is, the baselines can quickly adapt to new categories by updating only *a few* relevant hash embeddings and leaving the other parameters unmodified (ie. minimal forgetting). We take three variants: SlowAda only trains one epoch on the new data; MediumAda trains five epochs; FastAda trains 15 epochs. These baselines cover forgetting-adaptation trade-offs: FastAda is fast in adaptation but suffers forgetting; SlowAda is on the opposite end.

Table 1: Online learning results on TTD-streams. Adult, Bank, Mushroom, and Covertype are evaluated by average accuracy, the larger the better. Retail and MovieLens-32M use mean absolute error, lower the better. All results are multiplied by 100 except Retail for visual clarity. PHE achieves the best performance among all hash embedding-based methods.

| | Hash Embedding | | | | Collision-Free Embedding | |
|---|---|---|---|---|---|---|
| | SlowAda | MediumAda | FastAda | PHE (ours) | EE | P-EE |
| Adult ($\uparrow$) | $82.2 \pm 0.7$ | $74.8 \pm 4.5$ | $71.1 \pm 4.0$ | $\mathbf{84.1 \pm 0.2}$ | $84.2 \pm 0.0$ | $84.8 \pm 0.0$ |
| Bank ($\uparrow$) | $\mathbf{89.7 \pm 0.1}$ | $89.0 \pm 0.9$ | $86.9 \pm 1.6$ | $89.6 \pm 0.0$ | $90.0 \pm 0.0$ | $90.1 \pm 0.0$ |
| Mushroom ($\uparrow$) | $97.7 \pm 0.7$ | $97.9 \pm 0.5$ | $98.3 \pm 0.3$ | $\mathbf{98.8 \pm 0.0}$ | $98.8 \pm 0.0$ | $98.8 \pm 0.0$ |
| CoverType ($\uparrow$) | $63.5 \pm 0.5$ | $59.1 \pm 1.2$ | $55.3 \pm 1.2$ | $\mathbf{64.3 \pm 0.2}$ | $64.3 \pm 0.1$ | $64.0 \pm 0.4$ |
| Retail ($\downarrow$) | $49.1\pm82.9$ | $22.7\pm20.3$ | - | $\mathbf{3.0\pm0.2}$ | $3.7\pm0.1$ | $3.2\pm0.4$ |
| MovieLens ($\downarrow$) | $15.3\pm0.1$ | $15.1\pm0.1$ | $15.1\pm0.1$ | $\mathbf{14.7\pm0.0}$ | $15.1\pm0.0$ | $14.7\pm0.0$ |

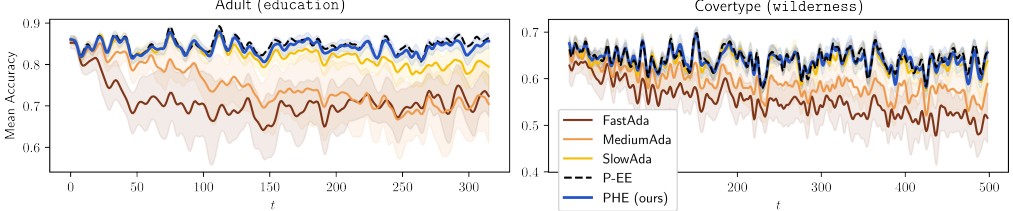

Figure 4: Online classification results on tabular data streams. The Ada results show a downward trend although there are no new items to learn, suggesting the deterministic hash embeddings suffer from forgetting during the learning. In contrast, the proposed PHE mitigates the forgetting issue and keeps performing as good as the upper-bound method P-EE. Other datasets in Fig. 6 in Supp. E.3 show similar conclusions. In the parentheses is the column whose items embeddings get updated.

Another compared method is an ideal method representing the upper bound–collision-free expandable embeddings (EE). In particular, EE dynamically initializes and updates an embedding from scratch when a new item is encountered. This baseline does not suffer from cross-item interference but may overfit to the most-recent observations. To mitigate the overfitting issue, we treated EE to be probabilistic and applied Bayesian online learning. We refer to as P-EE. (P-)EE is memory-inefficient as the memory scales linearly with vocabulary size and can grow unbounded, posing a challenge in large-scale applications (Tab. 2). Moreover, it is noticeably hard to implement (P-)EE, which requires to dynamically redefine the embedding layer upon observing new categories.

## 4.2 CLASSIFICATION IN TTD-STREAMS

Here we show the evolving vocabulary can be managed through PHE effectively on public datasets.

**Datasets.** We apply four public static tabular datasets that are available in UCI Machine Learning Repository: Adult, Bank, Mushroom, and Covertype. These datasets contain a mixture of discrete and continuous columns and are collected for classification problems in various domains. For stability, we normalized all continuous columns such that the value ranges from zero to one.

**Experimental setups.** We perform the classification tasks the original datasets provide. The generating data assumption is illustrated in Fig. 2b.2. Regarding model architecture, we concatenate all category embeddings as well as continuous features as input to a deterministic neural network, followed by a softmax activation function. We assume the targets follow categorical distribution.

To simulate the data-streaming setup, at each step we present a randomly sampled data mini-batch to the model and evaluate the online learning performance. We require only one column's item embeddings be updated, mimicking that column has a changing vocabulary. Besides, we initialize the model (both embeddings and neural network weights) with a separate random portion of the data.

**Results.** We reported the data-streaming online classification accuracy in Fig. 4. The facts that 1) any items seen during online learning have been learned at the initialization and that 2) the accuracy curves of Ada methods have a downward trend suggest hash embeddings suffers from forgetting. In fact, the forgetting is caused by *parameter interference* in hash embeddings: suppose items A and B share parameters in the hash embedding table, then updating A's embedding affect B's embedding.

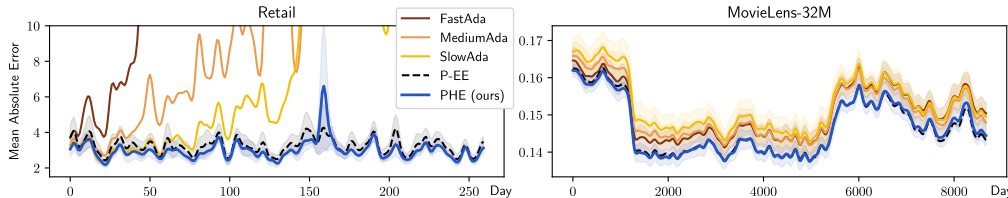

Figure 5: Experimental results of sequence modeling (left) and recommendation (right) on large-scale TTD-streams Retail and MovieLens-32M. For visibility, all curves are smoothed. It shows that our method PHE outperforms all deterministic hash embedding baselines (Fast/Medium/SlowAda) that are sensitive to their optimization hyperparameters. Moreover, it is remarkable to note that PHE performs similarly with the upper-bound collision-free P-EE baseline, especially considering PHE consumes only 2% and 4% of the size of P-EE (see Tab. 2). The initial performance gap at Day 0 on MovieLens is an artifact of smoothing; in fact, all methods have similar initial performance.

We further reported an overall averaged accuracy in Tab. 1. The results show that our proposed PHE performs similarly with the upper-bound collision-free embeddings (EE), and the gap between PHE and all other deterministic counterparts proves the effectiveness of PHE in online learning. Besides, PHE is more stable and has a smaller variance. The varying performances of the Ada baselines highlight the importance and sensitivity of hyperparameter tuning. In contrast, our method is a hyperparameter-free approach and the only demand is to train the model until convergence. Lastly, as summarized in Tab. 2, PHE consumes noticeably lower memory than P-EE.

### 4.3 MULTI-TASK SEQUENCE MODELING IN TTD-STREAMS

Sequence models, exploiting the temporal correlation among observations, predict a variable of interest based on the history.

**Datasets.** We apply a public large-scale time-stamped tabular dataset, Retail, an instance of TTD. A snippet of this dataset can be found in Tab. 4. This dataset records all online transactions between 01/12/2010 and 09/12/2011 in a retail store. There are over 4,000 products and over 540K time-stamped invoice records in total. The task is to predict the sales for each product shown in each invoice record given the product's historical sales.

**Experimental setups.** The Retail dataset is naturally a TTD-stream. We use the first three month data to initialize the model. Then we make predictions on a daily basis following the invoice timestamp. And at each step, we predict the sales quantity for each product on invoices based on their sale history. After that, we will receive the prediction error and use it to update the product embeddings. We use mean absolute errors as the evaluation metric. More details are in Supp. E.4.

We employ latent time variables to model correlations among neighboring transactions and hash embeddings to represent products. The latent time and product embeddings are independent and jointly account for the sales (see model assumption in Fig. 2b.1). Concretely, we use gated recurrent unit (GRU) (Chung et al., 2014) for both the generation and inference network. The network weights are frozen after the initial training.

**Results.** Fig. 5 shows the running performance (smoothed by a 1-D Gaussian filter): the Ada-family baselines favor shorter optimization time for Retail–FastAda explodes after 50 days. (The error bar is omitted as it is too large to be meaningful.) On the other hand, PHE has lower errors and is stable across all learning steps. Remarkably, on the average performance in Tab. 1, PHE significantly outperforms all baselines, including collision-free P-EE with only 2% memory usage. One possible reason is that P-EE initializes new embeddings from scratch and thus gets slow in warm-up, while PHE uses shared parameters from initial training.[5] Similar observations also occur in the continual learning setup (see Fig. 11 and Tab. 5 in Supp. E.4) and another large-scale recommendation task.

### 4.4 RECOMMENDATION IN TTD-STREAMS

TTD is a common in recommendation systems. For example, new users or movies reach a streaming service, the recommender needs to incorporate them and make recommendations.

---

[5]Another possible reason is that PHE provides a regularization mechanism (Tito Svenstrup et al., 2017).

Table 2: Number of parameters in the embedding module. Ratio is computed by dividing PHE by P-EE. The results show PHE consumes as low as $2\%$ of number of parameters (i.e. hardware memory) of P-EE, implying the memory-efficiency benefit of PHE. (See details in Supp. E.6.2.)

|  | Adult | Bank | Covertype | Mushroom | Retail | MovieLens-32M |
|---|---|---|---|---|---|---|
| PHE (ours) | 346 | 346 | 346 | 56 | 5014 | 460414 |
| P-EE | 3920 | 1760 | 1760 | 90 | 332760 | 11541320 |
| Compression Ratio | 0.09 | 0.2 | 0.2 | 0.62 | 0.02 | 0.04 |

**Datasets.** We apply the largest MovieLens-32m (Harper and Konstan, 2015) which contains 32 million ratings across over 87k movies and 200k users. These data were recorded between 1/9/1995 and 10/12/2023 for about 28 years. Each piece of data is a tuple of (userId, movieId, rating, timestamp), recording when and which rating a user gave a movie. Ratings ranges from 0 star to 5 star with half-star increments. This dataset is also an instance of TTD.

**Experimental setups.** We treat the recommendation problem as a rating prediction problem, where the task is predicting the rate a user gives to a movie. In implementation, ratings are normalized to $[0, 1]$ and are taken to be continuous albeit their increments are discrete. We simulate the experiment as in production – online prediction along the timestamp. We combine PHE and Neural Collaborative Filtering (He et al., 2017) as the backbone model. The model is pre-trained on the first five years of data and then perform predict-update online learning on a daily basis. We assume all ratings are IID conditioned on user, movie, and movie-genre embeddings. And we model user and movie embeddings through PHE while movie-genres are encoded as multi-hot embeddings. In this setup, both forgetting and adaptation in the hash embeddings are measured: the model should avoid forgetting for recurring users/movies and adapt for new users/movies. Prediction error is evaluated by mean absolute error.

**Results.** The results of all compared methods are shown in Fig. 5 and the memory efficiency of PHE is reported in the last column of Tab. 2. The curves in Fig. 5 are smoothed with a 1-D Gaussian filter. The initial performance gap at Day 0 is an artifact of smoothing, in fact, all methods have similar performance on Day 0 (see Fig. 7 in Supp. E.5). It shows that our method PHE outperforms all deterministic hash embedding baselines (Fast/Medium/SlowAda) that have various forgetting-adaptation trade-offs. Similarly with the Retail dataset, PHE also significantly outperforms the collision-free P-EE baseline. This is remarkable considering PHE consumes only 4% of the memory of P-EE (Tab. 2). EE, the deterministic counterpart of P-EE, has worse performance, showing Bayesian online learning effectively mitigates overfitting. [6]

### 4.5 ADDITIONAL RESULTS

We conducted additional experiments and presented the results in Supp. E.6. We showcased additional results beside Fig. 1 in Supp. E.6.1; demonstrated the **memory and hardware efficiency** of PHE in Supp. E.6.2 (also see Tab. 2); analyzed **adaptation and forgetting** separately in Supp. E.6.3; investigated classification and sequence modeling in classical **continual learning setup** in Supp. E.6.4; performed **ablation studies** on the hash size $B$ and number of hash functions $K$ in Supp. E.6.5.

### 5 CONCLUSIONS

In this work we unveiled the ineffectiveness of hash embedding in learning TTD with dynamic vocabulary. We addressed the problem of modeling TTD and presented probabilistic hash embeddings (PHE). We showcased PHE is a plug-in module for multiple ML models, allowing those models to learn TTD-streams. We derive a scalable inference algorithm to simultaneously learn the model parameters and infer the latent embeddings. Through Bayesian online learning, the model is able to adapt to new vocabularies without additional hyperparameters in a changing environment. We benchmark PHE on large-scale public datasets with TTD demonstrating the efficacy of PHE.

---

[6]An interesting observation: It turns out MovieLens made two major changes in their movie rating system around 2003 and 2014 (website), which is reflected in our online learning results – two sharp changes in Fig. 5.

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

## A    EVIDENCE LOWER BOUNDS

We denote the model parameters relevant to the generating process by $\theta := \{\theta_z, \theta_y\}$. To learn the model parameters, we maximize the marginal likelihood $p(\mathbf{y}_{\leq N}|\mathbf{x}_{\leq N}, \mathbf{h}_{\mathbf{s}_{\leq N}}; \theta)$. Directly optimizing this marginal likelihood with the Expectation-Maximization (EM) algorithm is intractable. Therefore, we jointly learn the model parameters $\theta$ and infer the variational posteriors of latent variables $\{E, \mathbf{z}_{<N}\}$ using the variational EM algorithm. That is, we maximize the evidence lower bound (ELBO) $\mathcal{L}(\theta, \lambda, \phi)$ with respect to model parameters $\theta$ and variational parameters $\{\lambda, \phi\}$.

### A.1    DERIVATION OF $\mathcal{L}(\theta, \lambda, \phi)$

Denote all the history $\{\mathbf{x}_{\leq i}, \mathbf{y}_{\leq i}, E_{\mathbf{h}_{\mathbf{s}_{\leq i}}}\}$ until row $i$ by $O_i$. We find the optimal parameters by maximizing the marginal evidence $p(\mathbf{y}_{\leq N}|\mathbf{x}_{\leq N}, \mathbf{h}_{\mathbf{s}_{\leq N}}; \theta)$. We take the logarithm of marginal evidence

$$\log p(\mathbf{y}_{\leq N}|\mathbf{x}_{\leq N}, \mathbf{h}_{\mathbf{s}_{\leq N}}; \theta) \tag{6}$$

$$= \log \int \frac{p(\mathbf{y}_{\leq N}, E|\mathbf{x}_{\leq N}, \mathbf{h}_{\mathbf{s}_{\leq N}}; \theta)}{q_\lambda(E)} q_\lambda(E) dE \tag{7}$$

$$\geq \mathbb{E}_{q_\lambda(E)}[\log p(\mathbf{y}_{\leq N}, E|\mathbf{x}_{\leq N}, \mathbf{h}_{\mathbf{s}_{\leq N}}; \theta) - \log q_\lambda(E)] \tag{8}$$

$$= \mathbb{E}_{q_\lambda(E)}[\log p(\mathbf{y}_{\leq N}|\mathbf{x}_{\leq N}, E_{\mathbf{h}_{\mathbf{s}_{\leq N}}}; \theta)] - D_{\mathrm{KL}}(q_\lambda(E)|p(E)) \tag{9}$$

where the inequality follows from Jensen's inequality. Next, we apply the same trick for another time to find a lower bound of Eq. (9). Specifically, we will find a tractable lower bound to the conditional likelihood $\log p(\mathbf{y}_{\leq N}|\mathbf{x}_{\leq N}, E_{\mathbf{h}_{\mathbf{s}_{\leq N}}}; \theta)$.

In the filtering setup, we note that $\log p(\mathbf{y}_{\leq N}|\mathbf{x}_{\leq N}, E_{\mathbf{h}_{\mathbf{s}_{\leq N}}}; \theta) = \sum_{i=1}^{N} \log p(\mathbf{y}_i|\mathbf{y}_{<i}, \mathbf{x}_{\leq i}, E_{\mathbf{h}_{\mathbf{s}_{\leq i}}}; \theta)$. If we can find a lower bound for each $\log p(\mathbf{y}_i|\mathbf{y}_{<i}, \mathbf{x}_{\leq i}, E_{\mathbf{h}_{\mathbf{s}_{\leq i}}}; \theta)$, then the summation of the lower bounds is also a valid lower bound for $\log p(\mathbf{y}_{\leq N}|\mathbf{x}_{\leq N}, E_{\mathbf{h}_{\mathbf{s}_{\leq N}}}; \theta)$.

$$\log p(\mathbf{y}_i|\mathbf{y}_{<i}, \mathbf{x}_{\leq i}, E_{\mathbf{h}_{\mathbf{s}_{\leq i}}}; \theta) \tag{10}$$

$$= \log \int \frac{p(\mathbf{y}_i, \mathbf{z}_i|\mathbf{y}_{<i}, \mathbf{x}_{\leq i}, E_{\mathbf{h}_{\mathbf{s}_{\leq i}}}; \theta)}{q_\phi(\mathbf{z}_i|O_i)} q_\phi(\mathbf{z}_i|O_i) d\mathbf{z}_i \tag{11}$$

$$\geq \mathbb{E}_{q_\phi(\mathbf{z}_i|O_i)}[\log p(\mathbf{y}_i|\mathbf{y}_{<i}, \mathbf{x}_{\leq i}, E_{\mathbf{h}_{\mathbf{s}_{\leq i}}}, \mathbf{z}_i; \theta)] - D_{\mathrm{KL}}(q_\phi(\mathbf{z}_i|O_i)|p(\mathbf{z}_i|O_{i-1})) \tag{12}$$

$$\geq \mathbb{E}_{q_\phi(\mathbf{z}_i|O_i)}[\log p(\mathbf{y}_i|\mathbf{y}_{<i}, \mathbf{x}_{\leq i}, E_{\mathbf{h}_{\mathbf{s}_{\leq i}}}, \mathbf{z}_i; \theta)] - \mathbb{E}_{q(\mathbf{z}_{i-1}|O_{i-1})} D_{\mathrm{KL}}(q_\phi(\mathbf{z}_i|O_i)|p(\mathbf{z}_i|\mathbf{z}_{i-1}; \theta_z)) \tag{13}$$

Eq. (12) to Eq. (13) follows from the following inequality:

$$D_{\mathrm{KL}}(q_\phi(\mathbf{z}_i|O_i)|p(\mathbf{z}_i|O_{i-1})) \leq \mathbb{E}_{q(\mathbf{z}_{i-1}|O_{i-1})} D_{\mathrm{KL}}(q_\phi(\mathbf{z}_i|O_i)|p(\mathbf{z}_i|\mathbf{z}_{i-1}; \theta_z)) \tag{14}$$

because

$$D_{\mathrm{KL}}(q_\phi(\mathbf{z}_i|O_i)|p(\mathbf{z}_i|O_{i-1})) \tag{15}$$

$$= \mathbb{E}_{q_\phi(\mathbf{z}_i|O_i)}[\log q_\phi(\mathbf{z}_i|O_i) - \log p(\mathbf{z}_i|O_{i-1})] \tag{16}$$

$$= \mathbb{E}_{q_\phi(\mathbf{z}_i|O_i)}\left[\log q_\phi(\mathbf{z}_i|O_i) - \log \mathbb{E}_{q(\mathbf{z}_{i-1}|O_{i-1})}[p(\mathbf{z}_i|\mathbf{z}_{i-1}; \theta_z)]\right] \tag{17}$$

$$\leq \mathbb{E}_{q_\phi(\mathbf{z}_i|O_i)}\left[\log q_\phi(\mathbf{z}_i|O_i) - \mathbb{E}_{q(\mathbf{z}_{i-1}|O_{i-1})}[\log p(\mathbf{z}_i|\mathbf{z}_{i-1}; \theta_z)]\right] \tag{18}$$

$$= \mathbb{E}_{q(\mathbf{z}_{i-1}|O_{i-1})q_\phi(\mathbf{z}_i|O_i)}[\log q_\phi(\mathbf{z}_i|O_i) - \log p(\mathbf{z}_i|\mathbf{z}_{i-1}; \theta_z)] \tag{19}$$

$$= \mathbb{E}_{q(\mathbf{z}_{i-1}|O_{i-1})} D_{\mathrm{KL}}(q_\phi(\mathbf{z}_i|O_i)|p(\mathbf{z}_i|\mathbf{z}_{i-1}; \theta_z)) \tag{20}$$

where Eq. (17) takes the Kalman filter prediction step.

Then Eq. (13) is the conditional ELBO $\mathcal{L}_i(\theta, \phi | E)$, i.e., Eq. (4). Plug Eq. (13) in Eq. (9), we have

$$\mathbb{E}_{q_\lambda(E)}[\log p(\mathbf{y}_{\leq N} | \mathbf{x}_{\leq N}, E_{\mathbf{h}_{s_{\leq N}}}; \theta)] - D_{\mathrm{KL}}(q_\lambda(E) | p(E)) \tag{21}$$

$$\geq \mathbb{E}_{q_\lambda(E)} \left[ \sum_{i=1}^N \mathcal{L}_i(\theta, \phi | E) \right] - D_{\mathrm{KL}}(q_\lambda(E) | p(E)) \tag{22}$$

which is our objective function $\mathcal{L}(\theta, \phi, \lambda)$ (Eq. (3)).

## A.2 $\mathcal{L}(\theta, \phi, \lambda)$ AS A VARIATIONAL EM ALGORITHM

*Why is maximizing $\mathcal{L}(\theta, \phi, \lambda)$ a meaningful objective as a variational expectation-maximization algorithm?* We start with a general latent variable model $p_\theta(x, z) = p(z)p_\theta(x|z)$ and infer the posterior $p_\theta(z|x)$.

$$D_{\mathrm{KL}}(q_\lambda(z) | p_\theta(z|x))$$
$$:= \mathbb{E}_{q_\lambda(z)}[\log q_\lambda(z) - \log p_\theta(z|x)]$$
$$= \mathbb{E}_{q_\lambda(z)}[\log q_\lambda(z) - \log p_\theta(x, z) + \log p_\theta(x)]$$
$$= -\mathcal{L}(\lambda, \theta) + \log p_\theta(x)$$

Re-ordering the equation yields

$$\mathcal{L}(\lambda, \theta) = \log p_\theta(x) - D_{\mathrm{KL}}(q_\lambda(z) | p_\theta(z|x)),$$

which shows that maximizing the ELBO $\mathcal{L}(\lambda, \theta)$ is equivalent to both maximizing the marginal likelihood $p_\theta(x)$ and minimizing the inference gap $D_{\mathrm{KL}}(q_\lambda(z) | p_\theta(z|x))$.

Then, with the same procedure as above, two facts follow: 1) maximizing $\mathcal{L}_i(\theta, \phi | E)$ is equivalent to maximizing the conditional likelihood $\log p(\mathbf{y}_i | \mathbf{y}_{<i}, \mathbf{x}_{\leq i}, E_{\mathbf{h}_{s_{\leq i}}}; \theta)$ and minimizing the inference gap $D_{\mathrm{KL}}(q_\phi(\mathbf{z}_i | O_i) | p(\mathbf{z}_i | O_i; \theta))$ simultaneously; 2) maximizing Eq. (9) is equivalent to maximizing $\log p(\mathbf{y}_{\leq N} | \mathbf{x}_{\leq N}, \mathbf{h}_{s_{\leq N}}; \theta)$ and minimizing the inference gap $D_{\mathrm{KL}}(q_\lambda(E) | p(E | \mathbf{y}_{\leq N}, \mathbf{x}_{\leq N}, \mathbf{h}_{s_{\leq N}}; \theta))$ simultaneously. Since maximizing $\mathcal{L}(\theta, \phi, \lambda)$ optimizes both $\mathcal{L}_i(\theta, \phi | E)$ and Eq. (9), we conclude our objective function will optimize all the mentioned aspects above.

## A.3 DERIVATION OF $\mathcal{L}^{(1)}(\lambda; \theta^*, \lambda_0^*, \phi^*)$

**Derivation of Eq. (5).** We only adapt the probabilistic hash embedding $E$. Similar to Bayesian online learning where the previous posterior is used as the new prior, we use the previous approximate posterior $q_{\lambda_0^*}(E)$ as the new prior for dataset $\mathcal{D}_1$ and fix all the other model parameters $\theta^*, \phi^*$. The derivation is the same as the one for Eq. (9) except we replace $p(E)$ with $q_{\lambda_0^*}(E)$. We only update $\lambda$ to acquire the new posterior in the optimization.

## B RELATED WORK

| | D0 | D1 | D2 | D3 | D4 |
|---|---|---|---|---|---|
| Changing vocabulary | | | ✔ | ✔ | ✔ |
| Timestamped | | ✔ | | ✔ | ✔ |
| Multi-task | | | | | ✔ |

Table 3: Tabular datasets can be categorized into five categories ($D0 - D4$) based on combinations of three characteristics, i.e., whether their categorical feature vocabulary dynamically expands over time, whether they contain a specific timestamp column, and whether their nature is multi-task. For example, datasets without all these characteristics are considered static ($D0$). While existing works mainly consider $D0$ and $D1$, PHE fits all dataset types ($D0 - D4$) and specifically highlights the unique applicability for dynamic and temporal tabular data types ($D1 - D4$).

We extend the discussion in Sec. 2 and survey more related works. In a nutshell, our PHE applies to all tabular data types in Tab. 3 (i.e., $D0 - D4$) while existing works are targeted to $D0$ or $D1$.

Our work deals with multi-task dynamic temporal tabular data. Our method has two major components: the probabilistic hash embeddings that learn categorical feature representations and the latent variable model for multi-task temporal tabular data. Next, we discuss the main related works.

**Hash features.** PHE is motivated by hashing tricks. Weinberger et al. (2009) proposed to use one hash function to map categorical features to a one-hot hash embedding of length $B$, which is the bucket size. The drawback is the embedding size is too large because there is only one hash function and that requires a large bucket size $B$ to get rid of collision. Bloom Embeddings (Serrà and Karatzoglou, 2017) is based on Bloom filters and achieves efficient computation while maintaining a compact model size. Other previous work on using hashing tricks to generate features focuses on using a smaller number of embedding-related parameters to achieve the same performance as using one-hot encoding. Hash embeddings or unified embeddings (Tito Svenstrup et al., 2017; Cheng et al., 2023) use a shared embedding table for all categorical features and multiple hashing functions as indices of the embedding table, reducing the possibility of collision. Hash embeddings are designed for stationary vocabularies, emphasizing small parameter sizes. We generalize hash embeddings to a probabilistic version that enables us to learn changing vocabularies via Bayesian online learning. Composition Embeddings (Shi et al., 2020a) use multiple hash embedding tables; in contrast, PHE uses one shared embedding table, further reducing the memory cost. Wolpertinger (Dulac-Arnold et al., 2015) and Deep Hash embedding (Kang et al., 2021) use a deep neural network to encode features into real-valued embeddings. In a changing vocabulary setup, the drawback is the need to modify the whole neural network to incorporate new string features, even though there is only one new feature. Different from previous works, our method emphasizes the usage of hash embeddings in dynamic tabular data with changing vocabularies. In the meantime, the model architecture remains stable, and only partial parameter updates are required.

**Generative models for tabular data.** Recent research on generative models of non-temporal tabular data focuses on modeling multi-modality or heterogeneity but overlooks the sustainable representations for dynamically expanded vocabularies. These works rely on one-hot encoding for categorical features. Xu et al. (2019) learns VAE and GAN-based tabular data generator while conditioning on discrete categorical features. Later works rely on GAN to design tabular data generators (Liu et al., 2023b; Zhao et al., 2021). Kotelnikov et al. (2023) extend diffusion models to tabular data.

**Temporal tabular data models.** To our knowledge, there isn't a sequence model designed for multi-task temporal tabular data, although some previous works have the potential to extend to tabular data. PHE extends Deep Kalman Filters (Krishnan et al., 2015) to be applicable for multi-task, temporal, and dynamic tabular data, while the original Deep Kalman Filters do not explicitly consider the multi-task and dynamic vocabulary property of the tabular data. Girin et al. (2021) survey a list of latent variable sequence models that are possible to be extended to tabular data, although most of them are designed for speech or video data.

**Others.** The setup of learning dynamic tabular data with changing vocabularies shares the similarity to continual learning and Bayesian online learning (Kirkpatrick et al., 2017; Wang et al., 2023; Zenke et al., 2017; Nguyen et al., 2018; Li et al., 2021), but the difference is our formulation is a novel dictionary- or vocabulary-incremental setup for tabular data. Besides, Kireev et al. (2023) learn transferable robust embeddings for categorical features. Yin et al. (2020); Iida et al. (2021) design objective functions for representation learning on tabular data using large-language models. Arik and Pfister (2021) and Huang et al. (2020) use the one-hot encoder to learn categorical feature embeddings before input to a transformer module.

**Discussions on alternative designs and shortcomings.** We acknowledge that alternative solutions may exist, e.g., encoding string features with a character-level recurrent neural network or using a popularity-based token-level one-hot encoder. In our considered aspects, for example, long-tailed data distributions are commonly seen in applications, probabilistic hash embedding stands out with simplility and continual learning capability. Hash features (Weinberger et al., 2009; Cheng et al., 2023) is memory inefficient. Incremental one-hot embeddings are also inefficient for dynamic tabular data, because the model parameters expand unbounded, resulting in storage inefficient and slow computation. Deep hash embedding (Kang et al., 2021) and other methods in the same fashion are

Table 4: A tabular data snippet from the Retail dataset. The columns are either categorical, numeric, or timestamp. The rows corresponds to sale records. `StockCode` stores product ID. `Quantity` stores the sales. "?" denotes missing values. The task is to predict the sales for each product.

| StockCode | Date | UnitPrice | CustomerID | Country | Quantity |
|---|---|---|---|---|---|
| 85123A | 2010-12-01 08:26:00 | 2.55 | 17850 | United Kingdom | 6 |
| 84406B | 2010-12-01 08:26:00 | 2.75 | 17850 | United Kingdom | 6 |
| 21724 | 2010-12-01 08:45:00 | 0.85 | 12583 | France | 12 |
| 21791 | 2010-12-01 10:03:00 | 1.25 | 12431 | Australia | 12 |
| 22139 | 2010-12-01 11:52:00 | 0.55 | ? | United Kingdom | 56 |

computationally inefficient. One needs to adapt the whole neural network even when adding one new category. In contrast, one only needs to adapt the corresponding embeddings in probabilistic hash embedding.

**Handling hashing value collisions.** Collision of hash values could happen among popular, important categories. To address this issue, we can select the desired hash functions that avoid important collisions before applying the hash functions. In addition, users come and go fast, and collisions may become unimportant over time.

## C   AN EXAMPLE TTD

We will explain the concepts related to this work through an example tabular data snippet (Tab. 4). Tabular data contains two dimensions–rows and columns. Any stored information can be located by specifying the row and column indices. We can classify columns into three types: *categorical*, *numeric*, and *timestamp*. A categorical column represents a discrete nominal feature, usually recorded in text strings and therefore hashable; A numeric column corresponds to a numeric feature, usually represented by float or integer values; and a timestamp column records the timestamp when a row is created. For instance, in Tab. 4, there are six columns, among which `StockCode`, `CustomerID`, and `Country` are categorical columns, `UnitPrice` and `Quantity` are numeric columns, and `Date` is a timestamp column. Some columns are of particular interest and one may want to predict those based on others. We refer to those columns as *predicted columns*. Predicted columns can be either categorical or numeric, depending on task requirements. Rows with similar timestamps usually exhibit correlations. But these correlations may change over time.

Some tabular data is multi-task-oriented. For example, in Tab. 4, one may be interested in predicting future selling quantity based on historical transactions for each product. In this case, different product IDs in `StockCode` suggest different tasks. We refer to the categorical columns consisting of task identifiers as *global columns* and other categorical columns as *local columns*. We express this type of tabular data *multi-task*. Each task may have specific column relationships.

All unique items in a categorical column constitute its *vocabulary*. When new items join into the column, we say it has a *changing* or *dynamic vocabulary*. [7] When any changes happen in the above three aspects for a table, we say it is *temporal tabular data* (TTD).

## D   A SIMPLIFIED MODEL TO UNDERSTAND WHY PHE IS SUPERIOR TO DHE

In this section, we consider a simple linear Gaussian model that we can analyze in closed form to illustrate why having deterministic hash embeddings that are updated in an online fashion is prone to *forgetting*. The crux of our calculations is the fact that distinct categorical items share representations due to partial hash collisions. Thus, when trained online, the shared features shows a bias to work well for the categorical item that was most recently seen, rather than be optimized for the overall data distribution seen so far, leading to the forgetting behaviour. However, we will show that Bayesian hash embedding does not suffer this, because it is well known that if exact online posterior can be

---

[7]We assume the tabular structure is fixed, i.e., the number of columns, column names, and types are fixed. We also assume categorical features are single-valued. But our work is compatible with multi-valued features.

computed (which in our linear Gaussian setup is easy to do), the online posterior is identical to the offline one.[8]

**A simple linear-Gaussian model**

Consider a simple situation of regression with input variable $X \in \{0, 1\}$ taking one of two categorical values and the target $Y \in \mathbb{R}$ is real-valued. The conditional distribution of $Y$ is a gaussian distribution with the mean being 1 when $X = 0$ and mean being $-1$ when $X = 1$. We further assume that the variance of $Y$ is $\sigma^2 \approx 0$ is tiny, In notation terms, the true distribution of $Y|X = 0 \sim \mathcal{N}(1, \sigma^2)$, while the distribution of $Y|X = 1 \sim \mathcal{N}(-1, \sigma^2)$, where $\sigma$ is a fixed and small. We do not specify the distribution of the covariate $X$ just yet and defer that to the sequel.

**The predictive model based on hash embedding**

Given labeled data $(X, Y)$, we aim to learn a predictor $f(X)$ that predicts $Y$ given $X$. To build the predictor we use a simple hash embedding model. Specifically, we assume that the predictor $f(\cdot)$ is parameterized by a $3 \times 1$ embedding matrix $E$. Although technically this is a vector, we still denote it as an 'embedding matrix' to be consistent with the rest of the exposition. Denote by $e^{(0)}, e^{(1)}$ and $e^{(2)}$ as the three rows of this matrix which are the 'embedding vectors' of the three hash values. Thus, in the notation of our model, this embedding matrix is made of $B = 3$ buckets with the dimension $d = 1$. The model $f(\cdot)$ uses two hash functions $h_i(\cdot) : \{0, 1\} \to \{0, 1, 2, \}$ to map the categorical variable $X$ into a hash value. Without loss of generality, we assume that $h_1(0) = 0, h_2(0) = 1, h_1(1) = 1, h_2(1) = 2$. Given this, the predictive model $f(X) := e^{(h_1(X))} + e^{(h_2(X))}$ is a simple linear sum of the two hash embedding of the input based on the two hash functions $h_1(\cdot)$ and $h_2(\cdot)$. This is a simple example of the general class of models where the predictor $Y$ is a linear function of the embedding vectors of the categorical input $X$ computed using the different hash functions. Although simple, this example illustrates the phenomenon that emerges of learning categorical variables in an online fashion since the embedding vector $e^{(1)}$ influences both $X = 0$ through hash function $h_1(\cdot)$ and $X = 1$ through hash function $h_2(\cdot)$.

**An online interaction setting**

We consider the following online prediction protocol. At each time $t = 1, 2, \cdots$, the environment samples $X_t$ from a distribution over $\{0, 1\}$ and produces to the predictor. The predictor then predicts $\widehat{Y}_t := f_t(X_t)$ and is then shown the true label $Y \in \mathbb{R}$. The predictor incurs loss $l_t := \frac{1}{2}(Y_t - \widehat{Y}_t)^2$ and uses the observed $Y_t$ to update the predictor to $f_{t+1}(\cdot)$.

The only learnable parameters of the predictor is the embedding matrix $E$. Thus the predictor at time $t$ denoted by $f_t(\cdot)$ is parametrized by the state of the embedding matrix $E_t$ with its three rows denoted by $e_t^{(i)}$ for $i \in \{0, 1, 2\}$.

**Update the hash embedding matrix through Online Gradient Descent (OGD)**

In order to demonstrate that the hash embeddings can lead to forgetting, we will assume that they are updated through standard online gradient descent. Observe that at time $t$, if $X_t = 0$, then $\widehat{Y}_t = e_t^{(0)} + e_t^{(1)}$. The instantaneous loss at time $t$ is given by $l_t = \frac{1}{2}(\widehat{Y}_t - Y_t)^2$. Thus, the gradients $\frac{\partial l_t}{\partial e^{(0)}} = \frac{\partial l_t}{\partial e^{(1)}} = (e_t^{(0)} + e_t^{(1)} - Y_t)$, if $X_t = 0$. Thus, assuming that the embedding matrix $E_t$ is updated online using OGD at a fixed learning rate $\eta \in \mathbb{R}$ leads to the following update equations

$$e_{t+1}^{(0)} = \begin{cases} e_t^{(0)} - \eta((e_t^{(0)} + e_t^{(1)} - Y_t)), & X_t = 0 \\ e_t^{(0)}, & X_t = 1. \end{cases}$$

Similarly the update equations for the other two embedding vectors are as follows.

$$e_{t+1}^{(1)} = \begin{cases} e_t^{(1)} - \eta((e_t^{(0)} + e_t^{(1)} - Y_t)) & X_t = 0 \\ e_t^{(1)} - \eta((e_t^{(1)} + e_t^{(2)} - Y_t)) & X_t = 1 \end{cases}$$

---

[8]Note that this section has a slightly different notation from the main text, but the content is self-contained. Readers can also match the notation by noting $X := \mathbf{m}$ and the input variable value has $0 := \mathbf{m}_0$ and $1 := \mathbf{m}_1$.

$$e_{t+1}^{(2)} = \begin{cases} e_t^{(2)} & X_t = 0 \\ e_t^{(2)} - \eta((e_t^{(1)} + e_t^{(2)} - Y_t)) & X_t = 1 \end{cases}$$

These update equations for the embedding shows that $e^{(1)}$ which is shared for both $X = 0$ and $X = 1$ gets updated all the time, while $e^{(0)}$ is only updated if $X = 0$ and similarly $e^{(1)}$ is only updated if $X = 1$.

**A non-stationary distribution for the co-variates $X$**

Consider a setting where the first $N$ inputs consists of $X_t = 0$ for all $t \in \{1, \cdots, N\}$, followed by another $N$ inputs consisting of $X_t = 1$ for all $t \in \{N + 1, \cdots, 2N\}$. In these discussions we will assume $N$ is large enough and the learning rate $\eta$ is appropriately tuned to make the variance of the predictor to be small. If all the $2N$ samples were shown to a training algorithm, it could have (near) perfectly estimated the embedding matrix $\widehat{E}$, i.e., for a $X$ that is sampled from $\{0, 1\}$ that is equally likely (matching the training data distribution of equal number of 0 and 1), the expected excess loss will be arbitrarily small (assuming $N$ is sufficiently large). We will show in the calculations below that if instead the embedding matrix was learnt using OGD, even if $N$ is large enough, the learnt model at the end will have a constant excess risk when the test input $X$ is sampled with equal probability among $\{0, 1\}$.

**Analyzing the OGD update equations**

To see this, we make some simplifying assumptions. First is that $\sigma = 0$, i.e., conditioned on $X$, $Y$ is deterministic. Second is a symmetric starting point of $e_0^{(i)} = 0$ for all $i \in \{0, 1, 2\}$. It is easy to observe that both of these assumptions do not change the the observation we will make, but makes the exposition easier. Thus, at the end of the first $N$ samples, we will have $e_{N+1}^{(2)} = 0$ and $e_{N+1}^{(0)} = e_{N+1}^{(1)} \approx 1/2$. This follows as $N$ is large and the noise $\sigma$ is 0, thus leading OGD to converge to a local minima of the loss function. Any embedding matrix with $e^{(0)} + e^{(1)} = 1$ is a local-minimum of the loss function and thus at the end of time $N + 1$, OGD will result in $e_{N+1}^{(0)} + e_{N+1}^{(1)} \approx 1$. Since the initialization and the loss function is symmetric in the arguements $e_t^{(0)} = e_t^{(1)}$ will hold for all $t \leq N$.

At time $t = N + 1$, the $N$ observed samples corresponds to $X = 0$. Thus, the prediction error for $X = 0$ by this learnt model $\widehat{f}_{N+1}(X)$ is small, i.e., the excess risk $(f_{N+1}(X) - 1)^2 \approx 0$.

Now consider the times $t = N + 1$ till $t = 2N$. During this period, the gradients will not impact $e^{(0)}$, i.e., $e_{N+1}^{(0)} = e_{2N+1}^{(0)} \approx 1/2$. However, $e^{(2)}$ and $e^{(3)}$ are no longer symmetric. But one can work out the recursion for their evolution since the gradients are the same.

In particular, for any time $t \in \{N + 1, \cdots, 2N\}$, the observed $X_t = 1$. Thus, the gradient of $e^{(1)}$ and $e^{(2)}$ at all times $t \in \{N + 1, \cdots, 2N\}$ is the equal to $(e_t^{(1)} + e_t^{(2)} + 1)$. Thus, under the OGD update equations, for all times $t \in \{N + 1, \cdots, 2N\}$, the equality $e_{t+1}^{(1)} - e_{t+1}^{(2)} = e_t^{(1)} - e_t^{(2)}$, holds. Since at time $N + 1$, we have $e_{N+1}^{(1)} \approx 1/2$ and $e_{N+1}^{(2)} = 0$, we have that $e_{2N+1}^{(1)} - e_{2N+1}^{(2)} \approx 1/2$. On the other hand, if $N$ is large, we know that OGD will converge to a local minima, i.e., $e_{2N+1}^{(1)} + e_{2N+1}^{(2)} \approx -1$. These two equations in the variables $e_{2N+1}^{(1)}, e_{2N+1}^{(2)}$ gives $e_{2N+1}^{(1)} \approx -1/4$ and $e_{2N+1}^{(2)} \approx -3/4$.

**Concluding that the updates leads to forgetting the representation for $X = 0$**

Thus at the end at time $2N + 1$, after having seen the first $N$ samples of $X = 0$ and the last $N$ samples of $X = 1$, the predictor is such that $\widehat{f}_{2N+1}(0) \approx 1/4$ and $\widehat{f}_{2N+1}(1) \approx -1$. However, note that the true label when $X = 0$ is 1 while when $X = 1$ is $-1$. Thus, the predictor $\widehat{f}_{2N+1}(\cdot)$ has near zero prediction error when $X = 1$. However, when $X = 0$, the loss given by $(\widehat{f}_{2N+1}(0) - Y)^2 \approx (1/4 - 1)^2 \approx 9/16$ is a constant.

This shows the discrepancy between a model trained offline using all the $2N$ samples and the model trained online where the first $N$ samples all correspond to $X = 0$ and the last $N$ samples correspond to $X = 1$. The offline model will converge to a local minima in which the prediction error for both

$X = 0$ and $X = 1$ will be small, while the online model converges to a solution where the prediction error for the categorical variable that was not seen recently is high.

**Arguing that online Bayesian model does not lead to forgetting**

A Bayesian method to 'learn' the embedding matrix is to posit a prior distribution $p(E)$ for the emebedding matrix and then given the data $\mathbf{X}$ compute the posterior distribution $p(E|\mathbf{X})$. We will say that the Bayesian learning does not forget, if the posterior distribution computed based on all the $2N$ samples $(X_1, Y_1), \cdots, (X_{2N}, Y_{2N})$ shown up-front matches the posterior distribution computed in an online fashion. However, from classical results in online Bayesian learning, it is well known that if one can compute the exact posterior $p(E|X_1, \cdots, X_t)$ at all times $t$, then the posterior at time $2N$ is identical to the one that an offline algorithm would have computed had it seen all the $2N$ samples at once. Thus, if the exact posterior can be computed at each time, then there is no forgetting in the Bayesian mechanism.

Thus in this section, we showed through a simple linear-gaussian model, that online updating of hash embedding matrix leads to forgetting while a bayesian updating of the embedding matrix does not lead to forgetting. In order to demonstrate this, we defined forgetting to not occur if the model learnt at the end of seeing each online sample one by one is close to the model learnt had all the samples been available up-front. Further, we show in experiments that this insight holds even in more complex scenarios where exact Bayesian posterior cannot be computed, but only an approximation through variational inference can be done.

# E EXPERIMENTAL DETAILS

## E.1 AN EFFICIENT EMBEDDING FETCH SCHEMES

When implementing the hash embedding fetching module, there are two available schemes: scheme one is first to sample a whole hash table $E$ and then fetch the corresponding embeddings $E_{\mathbf{h_s}}$ (as Eq. (23)); scheme two is first to fetch the distribution $p(E_{\mathbf{h_s}})$ and then sample $E_{\mathbf{h_s}}$ (as Eq. (24)).

$$p(\mathbf{x}|\mathbf{s}) = p(\mathbf{x}|\mathbf{h_s}) = \mathbb{E}_{p(E)}[p(\mathbf{x}|E, \mathbf{h_s})] \approx p(\mathbf{x}|E, \mathbf{h_s}) \tag{23}$$

$$= \mathbb{E}_{p(E_{\mathbf{h_s}})}[p(\mathbf{x}|E_{\mathbf{h_s}})] \approx p(\mathbf{x}|E_{\mathbf{h_s}}) \tag{24}$$

The two schemes lead to the same results, but scheme two is more memory-efficient as it does not need to sample the whole embedding table. Thus in practice, we apply Eq. (24).

## E.2 HARDWARE INFORMATION

We train and test our model on GPUs (RTX 5000) and use the deep learning framework PyTorch to enable efficient stochastic backpropagation. In all supervised learning experiments, the total elapsed wall time (training and testing) for PHE is less than half an hour, and the finetune baseline runs slightly faster. In the sequence modeling experiments, PHE runs about one hour since Retail is a large dataset and has over 500k records. In the recommendation experiments, it takes about two hours for all methods.

## E.3 DETAILS FOR CLASSIFICATION IN TTD

The four public datasets all can be found online: Adult[9], Bank[10], Mushroom[11], and Covertype[12]. Specifically, Adult has 14 columns and 48,842 rows containing demographic information. The task is to predict whether or not a person makes over $50K a year; Bank has 16 columns and 45,211 rows to predict if a client will subscribe to a term deposit; In Mushroom, of 22 discrete columns and 8,124 rows, the goal is to predict whether a mushroom is poisonous; Covertype, involving 12 columns and 581,012 rows, is to predict which forest cover type a pixel in a satellite image belongs to.

---

[9] https://archive.ics.uci.edu/dataset/2/adult
[10] https://archive.ics.uci.edu/dataset/222/bank+marketing
[11] https://archive.ics.uci.edu/dataset/73/mushroom We also follow the recommendation and only use `odor` as the feature.
[12] https://archive.ics.uci.edu/dataset/31/covertype

Regarding model architecture, we concatenate all category embeddings as well as continuous features as input to a deterministic one-layer neural network, followed by a softmax activation function. For PHE and P-EE, we stress that only embeddings are probabilistic and neural network weights are deterministic. We use negative cross entropy as the objective function assuming the targets follow categorical distributions.

We apply the following criterion when selecting a categorical column to have a dynamic vocabulary. We select the column to be dynamic if the weights of the column features have large scales when fitting a logistic regression model on the outputs. Specifically, we first use one-hot encodings to represent categorical items, and then fit a logistic regression model on the targeted outcomes. Finally, we select a column to be incremental if its corresponding categorical features have large weights because the weights in linear regression models can be interpreted as feature importance. Following this procedure, we select `education`, `poutcome`, `odor`, and `wilderness` column for the four datasets respectively. See detailed group information in Fig. 9.

For the continual learning setup in Supp. E.6.4, we first randomly and evenly split the categorical features of the selected column into disjoint groups, then partition the original dataset according to the groups. Based on the column dictionary size, we split Adult/Bank/Mushroom/Covertype into five/four/four/four disjoint groups. We randomly split each group into training and testing subsets where the training subset takes two-thirds of the total data and the testing subset takes the remaining one-third. We sequentially fit the prediction model to each non-overlapped group. The goal is to have high accuracy for all groups after sequential updates. Therefore, after fitting the model on the current group's training data, we report the average accuracy on all previous groups' test data.

**Evidence lower bound.** We first present the objective function of the latent variable supervised learning model (Fig. 2(b)). Similarly to Eq. (9), we can derive the objective function as the evidence lower bound of $\sum_{i=1}^{N} \log p(\mathbf{y}_i | \mathbf{x}_i, \mathbf{h}_{\mathbf{s}_i}; \theta)$:

$$\mathcal{L}(\theta, \lambda) = \mathbb{E}_{q_\lambda(E)} \left[ \sum_{i=1}^{N} \log p(\mathbf{y}_i | \mathbf{x}_i, E_{\mathbf{h}_{\mathbf{s}_i}}; \theta) \right] - D_{\mathrm{KL}}(q_\lambda(E) | p(E)) \tag{25}$$

For online adaptation to dataset $\mathcal{D}_1$ of size $N_1$, we fix the classifier parameters and only adapt the hash embedding table $E$. Denote the pre-trained parameters by $\theta^*$ and $\lambda_0^*$. Treat the previous posterior $q_{\lambda_0^*}(E)$ as the current prior, we can write down the objective function

$$\mathcal{L}^{(1)}(\lambda; \theta^*, \lambda_0^*) = \mathbb{E}_{q_\lambda(E)} \left[ \sum_{i=1}^{N_1} \log p(\mathbf{y}_i | \mathbf{x}_i, E_{\mathbf{h}_{\mathbf{s}_i}}; \theta^*) \right] - D_{\mathrm{KL}}(q_\lambda(E) | q_{\lambda_0^*}(E)) \tag{26}$$

**Implementation details and hyperparameters.** We implement the aggregation function $g$ as a weighted sum where the weights are parameters of $g$. Specifically, we have another random table $W \in \mathbb{R}^{P \times K}$ whose distribution is $p(W)$ and a hash function $h^{(W)} : \mathcal{S} \to \mathbb{N}_{<P}$ such that $h_{\mathbf{s}}^{(W)}$ indexes the rows of $W$, noted by $W_{h_{\mathbf{s}}^{(W)}} \in \mathbb{R}^K$. $W_{h_{\mathbf{s}}^{(W)}}$ serves as the weights for the $K$ hash embeddings (see Fig. 2a). Then $g(E_{h_{\mathbf{s}}^{(1)}}, \ldots, E_{h_{\mathbf{s}}^{(K)}}) = \sum_{k=1}^{K} W_{h_{\mathbf{s}}^{(W)}}^k E_{h_{\mathbf{s}}^{(k)}}$ where $W_{h_{\mathbf{s}}^{(W)}}^k$ is the $k$th value of vector $W_{h_{\mathbf{s}}^{(W)}}$. During inference, we infer the posteriors of both $E$ and $W$.

For all tabular datasets except Mushroom, we set $B = 7, K = 3, d = 20, P = 11$ (whose supported dictionary size is $P \times B^K = 3773$, which is ten times larger than the vocabulary size of the Adult dataset). We tried these values on Adult when setting the group size to be one (i.e., the static supervised learning setup) and found the resulting accuracy (about 84%) is comparable to the public results on this dataset[13]. We then use this same parameter setup on all other tabular data supervised learning experiments. For Mushroom, we use a much smaller model size and set $B = 5, K = 3, d = 5, P = 1$, because only one feature is used in the experiment.

**Optimization.** We use Adam stochastic optimization with a learning rate of 0.01 and a minibatch size of 128 in all experiments for both our method and baselines. For other hyperparameters of Adam, we apply the default values recommended in the PyTorch framework. When selecting these values, we fixed the minibatch size 128 and searched the learning rate (0.001, 0.005, 0.01, 0.05, 0.1) on the

---

[13]See the baseline model performance in `https://archive.ics.uci.edu/dataset/2/adult`

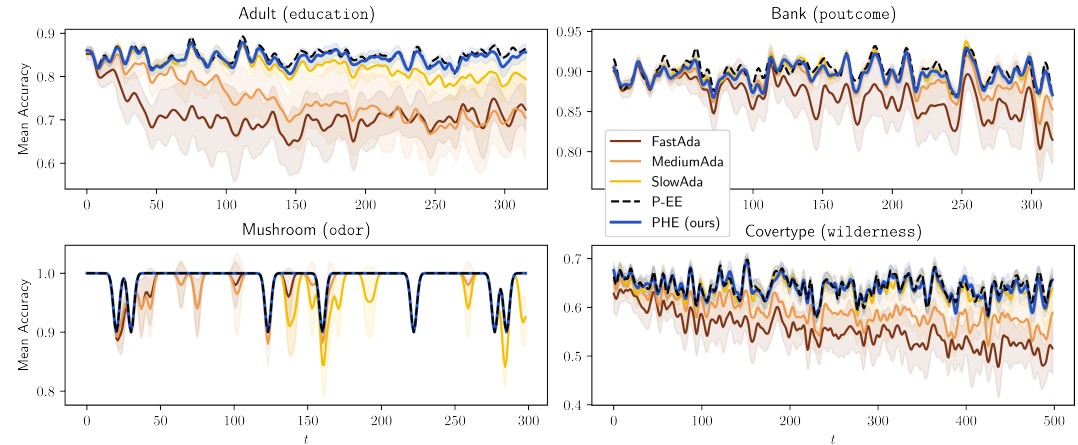

Figure 6: Results of online classification on all tabular data.

Adult dataset. We found the learning rate 0.01 leads to relatively fast and stable convergence. Then we apply the same values on all other datasets. For the first group training, we train PHE 100 epochs; for the remaining groups, we train PHE 15 epochs as we only need to update the hash embedding table $E$. Note that on every group, we train PHE until convergence.

**Evaluation metric.** We use accuracy as an evaluation metric. As we sequentially adapt the model on each vocabulary group's training set and test the model on the test set, we have running accuracies on each group.

**Additional results.** We add all datasets' online learning results in Fig. 6.

### E.4 DETAILS FOR MULTI-TASK SEQUENCE MODELING IN TTD

**Datasets.** We use the Retail dataset[14] as a multi-task TTD to demonstrate PHE. The dataset involves over 4,000 products indicated by StockCode column and the corresponding sale quantities represented by quantity column with invoice timestamps. We treat quantity as a time series and then track quantity for all 4,000 selling goods over time in a filtering setup. Prediction for each piece of product is regarded as one task and there are over 4,000 tasks in total. The task is to predict the sales quantity for the product shown in each invoice record given the product's previous sales.

For the continual learning setting in Supp. E.6.4, we treat all transactions as occurring at even time intervals. For each task, we randomly split the training and testing set with a ratio of 2:1. To get multi-tasks in a dynamic setting, we treat StockCode as the task identifier and evenly partition the products in StockCode into ten disjoint groups where each group involves about 400 goods, i.e., 400 new tasks. Correspondingly, the original dataset is converted into a task-incremental dataset where each task refers to predicting sale quantities (i.e., taking Quantity column values as $\mathbf{y}$) for one product, indicated by StockCode column. We normalize the UnitPrice column into the range $[0, 1]$ and do not use the Description column. We also drop cancellation transactions that have Quantity values smaller than zero. Therefore, we refer to StockCode as $\mathbf{u}$, Quantity as $\mathbf{y}$, UnitPrice as $\mathbf{x}$, {Country, CustomerId} as $\mathbf{m}$, and InvoiceDate as $t$.

**Evidence lower bound.** We assume the sales quantity follows Poisson distribution, consequently using the Poisson likelihood. As mentioned in the main paper, we use Eq. (3) to fit the first task and use Eq. (5) to fit the remaining tasks.

**Implementation details and hyperparameters.** We also implement a weighted aggregation function $g$ as above in the supervised learning setup. We did not try out different hyperparameter settings and directly set $B = 109, K = 3, d = 20, P = 109$ as these values can already support a large vocabulary (of size $P \times B^K$). We apply the same values to both PHE and the baselines.

---

[14]http://archive.ics.uci.edu/dataset/352/online+retail

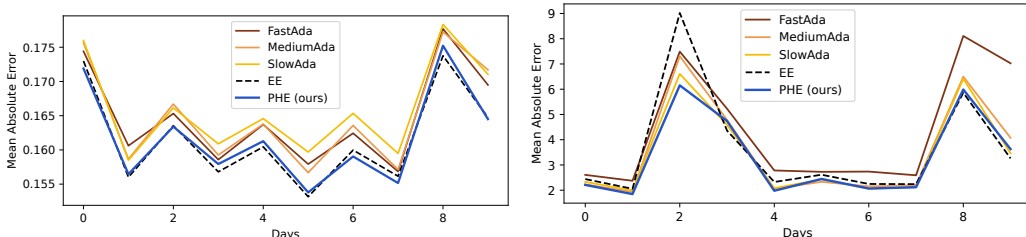

Figure 7: First ten day results of data-streaming movie recommendation and sales quantity sequence modeling.

**Optimization.** We use Adam stochastic optimization with the same learning rate of 0.005 and the same minibatch size of 128 as in supervised learning experiments. For other hyperparameters of Adam, we apply the default values recommended in the PyTorch framework. For the first task training, we train PHE 15 epochs; for the remaining tasks, we train PHE 5 epochs. Note that on every task, the epochs used are enough to train PHE until convergence.

**Evaluation metric.** We also evaluate the performance by the cumulative averages of errors. For each product, we use the first nine observations to predict the 10th observation and measure the absolute error on the 10th observation. Then, the average of all such absolute errors is the performance of this product. Since one group contains about 4,000 products, we further average each product's performance as the group's performance. Specifically, we have a prediction model that has a Poisson likelihood $p(\mathbf{y}_t|\mathbf{y}_{t-9:t-1}, \mathbf{x}_{t-9:t}, \mathbf{h}_{\mathbf{m}_{t-9:t}}, \mathbf{h}_{\mathbf{u}})$. We predict $\hat{\mathbf{y}}_t = \mathbb{E}[\mathbf{y}_t|\mathbf{y}_{t-9:t-1}, \mathbf{x}_{t-9:t}, \mathbf{h}_{\mathbf{m}_{t-9:t}}, \mathbf{h}_{\mathbf{u}}]$ as the mean value and then measure the absolute error between the ground-truth value $|\mathbf{y}_t - \hat{\mathbf{y}}_t|$.

For the continual learning setup, after learning group $t$, we can evaluate the performance of all previous and current groups, denoted by $R_{t,\leq t}$. We refer to the cumulative mean absolute error $\bar{R}_t = \sum_{a=1}^{t} R_{t,a}/t$ at group $t$ as the performance at $t$. We report $\bar{R}_t$ as a function of group numbers in Fig. 11. We report $\bar{R}_T$ after learning the final group $T$ in Tab. 5.

**Additional results.** Because we smoothed the results with a 1-D Gaussian filter in the main paper, we provide the first ten days' result without smoothing in Fig. 7.

### E.5 DETAILS FOR RECOMMENDATION IN TTD

Beside the first five years, this up-to-date and largest MovieLens dataset [15] have 8688 days (time steps) with possibly no records on some days. Note after pre-training, all model parameters are fixed except the hash embeddings. Regarding the likelihood function, we assume the rating follows Gaussian distribution.

We randomly split the data into a validation (20%) and a test set (80%). We searched the learning rate, batch size, neural network size, and likelihood scale on the validation set and reported final results on the test set. PHE, EE, and FastAda train the hash embeddings for 5 epochs per time step while MediumAda trains 2 epochs and SlowAda trains 1 epoch.

We also use the mean absolute error as the evaluation metric.

### E.6 ADDITIONAL RESULTS

### E.6.1 MORE MOTIVATION EXAMPLES

We report additional results in Fig. 8 as a complement to Fig. 1 in the main paper. Fig. 8 provides more evidence for the motivation of our work. For tabular data in a dynamic setting, not including the newly created categorical feature values in the prediction model will lead to a performance drop. Therefore, an efficient way to incorporate the new categorical features is necessary to maintain the efficacy of a prediction model. The "After update" performance in the plots demonstrates PHE is desirable for adapting to the new features. The splitting details are in Supps. E.3 and E.4 and Fig. 9.

---

[15]https://files.grouplens.org/datasets/movielens/ml-32m.zip

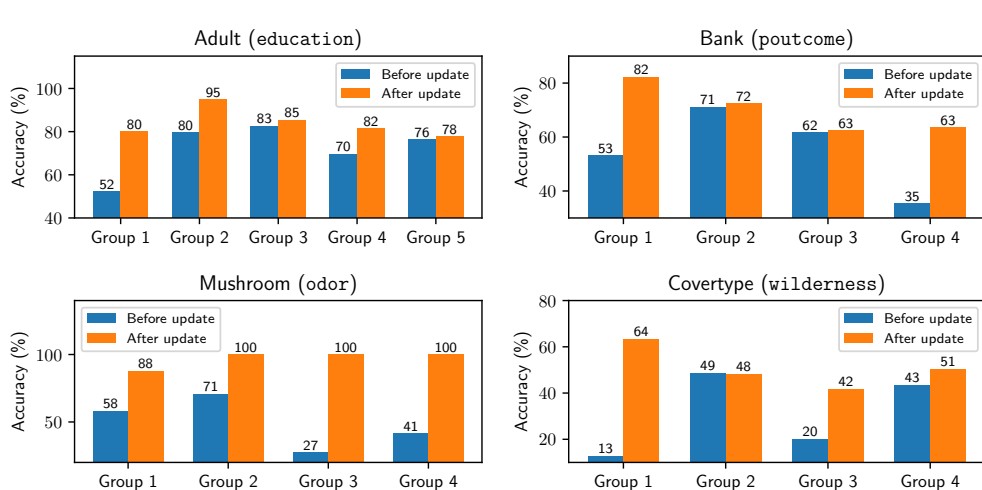

Figure 8: Adult dataset is randomly split into disjoint groups based on the `education` column. Groups arrive sequentially. We report results before and after the updates on the hash embeddings for each group to motivate the need to incorporate new groups into the model. Results are averaged on five independent runs with different random parameter initializations.

Adult

    Group 1 `(Preschool, 5th-6th, Bachelors)`

    Group 2 `(10th, 11th, 12th)`

    Group 3 `(7th-8th, HS-grad, Prof-school)`

    Group 4 `(9th, Assoc-voc, Doctorate)`

    Group 5 `(1st-4th, Masters, Some-college, Assoc-acdm)`

Mushroom

    Group 1 `(Musty, None)`

    Group 2 `(Anise, Almond)`

    Group 3 `(Spicy, Creosote)`

    Group 4 `(Foul, Fishy, Pungent)`

CoverType

    Group 1 `(A1)`

    Group 2 `(A3)`

    Group 3 `(A4)`

    Group 4 `(A2)`

Bank

    Group 1 `(Unknown)`

    Group 2 `(Failure)`

    Group 3 `(Other)`

    Group 4 `(Successs)`

Figure 9: Group information for continual classification tasks.

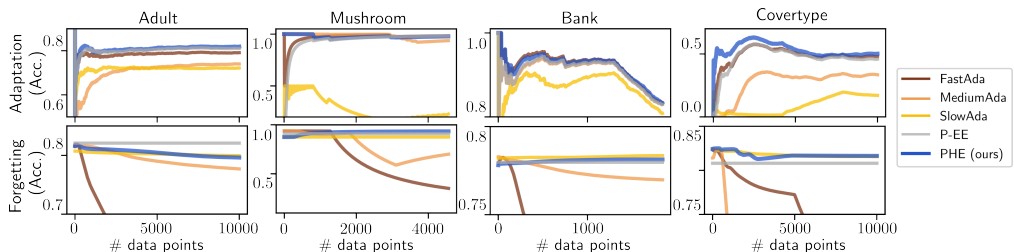

Figure 10: Comparision of online learning methods' adaptation and forgetting in a streaming online setup. Our PHE achieves similar performance with the collision-free P-EE on both metrics. Notably, SlowAda forgets the least but is slow in adaptation; FastAda is in the opposite regime.

### E.6.2 MEMORY EFFICIENCY

Memory efficiency of PHE can be seen from the number of parameters in the embedding module, which we summarized for both PHE and P-EE in Tab. 2. Note that P-EE sets the performance upper bound but its size scales linearly with the vocabulary size. The fact that PHE on all datasets achieves the same performance as P-EE illustrates PHE's impressive memory efficiency, especially considering PHE only consumes as low as $2\%$ memory of P-EE. Besides, being a unified embedding where all categorical columns share the same embedding table (Coleman et al., 2024), PHE is compatible with modern hardware and can benefit from the hardware acceleration.

We multiply each number by two because every parameter has its mean and variance. 20 is due to each embedding has 20 dimensions. For PHE, refer to implementation details (Supp. E) for the number of parameters ($B \times d + P \times K$). We compute the P-EE parameter size by $V \times d$ where $V$ is the vocabulary size.

### E.6.3 ADAPTATION AND FORGETTING ANALYSIS

**Adaptation and forgetting analysis.** We designed experiments to specifically measure the adaptation to new data and forgetting of old data. We split the data into two disjoint groups based on a random partition of one column's vocabulary. The model was initialized using the first group and online updated on the second group whose items are unseen in initialization. We let the data arrive one at a time. Adaptation is measured by the cumulative predictive accuracy of new datum and the forgetting by the accuracy of the first group's test data. Results in Fig. 10 show that our PHE has almost the best adaptation and forgetting performance on all four datasets. The P-EE while does not suffer forgetting, its adaptation to new categories is slow as each new embedding is initialized at random.

Regarding baselines, SlowAda uses a small learning rate (1e-4); MediumAda uses a medium learning rate (1e-3); FastAda uses a large learning rate (1e-2).

In Fig. 10, we compare on all four classification datasets used in the paper, our PHE against the four baselines. We observe from Fig. 10 that the SlowAda baseline with smaller LR (1e-4) leads to slower forgetting at the cost of slower adaptation, while larger LR (1e-2) has faster adaptation at the cost of faster forgetting (FastAda). Thus a data-stream dependent LR is needed for deterministic hash embeddings to trade off adaptation and forgetting. In contrast, our PHE has almost the best adaptation and forgetting performance on all four datasets due to the regularization from the posteriors. The EE while does not suffer forgetting as each category has a separate row in the embedding table, its adaptation to new categories is slow as each new embedding is initialized at random.

### E.6.4 CONTINUAL LEARNING

We also investigated classification and sequence modeling in the continual learning setup (Kirkpatrick et al., 2017), we split the dataset into disjoint groups based on a random partition of a selected column's vocabulary, assuming data distribution differs conditioned on each partition. This is similar to Supp. E.6.1. We then sequentially update the embeddings on each group's training data. After each group training, we evaluated the model performance on all previously seen groups' test data. The splitting details are in Supps. E.3 and E.4 and Fig. 9. While data-streaming setup aims to have good performance on the latest task, the goal of continual learning is to perform well on all groups after

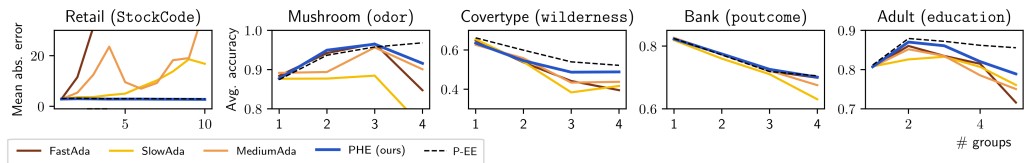

Figure 11: Cumulative average results. Column names in the parentheses are the ones made to have changing vocabulary and used to split groups. PHE is closest to the performance upper-bound P-EE.

Table 5: TTD batch continual learning performance of compared methdos. Adult, Bank, Mushroom, and Covertype are classification tasks and thus evaluated by average accuracy, which is larger the better. Retail is a regression task and we use the metric mean absolute error, lower the better.

|  | SlowAda | MediumAda | FastAda | P-EE (collision-free) | PHE (ours) |
|---|---|---|---|---|---|
| Adult | 76.1±1.8 | 75.0±4.7 | 71.6±3.1 | 85.6±0.1 | 78.9±3.0 |
| Bank | 63.0±4.0 | 67.5±4.5 | 69.9±1.2 | 70.5±0.7 | 70.1±1.4 |
| Mushroom | 75.5±7.6 | 90.1±8.6 | 84.7±12.3 | 96.8±0.0 | 91.6±7.6 |
| Covertype | 41.7±4.0 | 43.8±5.7 | 39.5±5.1 | 52.2±1.1 | 48.8±2.3 |
| Retail | 16.8±17.6 | 38.9±50.9 | - | 2.92±0.16 | 2.73±0.23 |

sequential training. Fig. 11 and Tab. 5 summarizes the results. Our PHE has the top performance among hash embedding methods.

### E.6.5    ABLATION STUDIES

**Justification of updating protocols.** We provided evidence on our updating protocols in Tab. 6, showing updating incremental column's embeddings as well as fixing other parameters has the best performance. Tab. 6 presents the accuracy of multiple updating schemes, justifying this updating protocol in use achieves both high accuracy and computational efficiency.

**The impact of potential hash collisions and the mitigation measures.**

We experimented on the large Retail dataset under the continual learning setup as in Supp. E.6.4. We varied the hyperparameters bucket size B and the number of hash functions K to control the potential number of hash collisions. In particular, we varied one hyperparameter when fixing the other.

We repeated each experiment five times with different random seeds. The tables below show the mean absolute errors (the lower the better) with standard deviation under each hyperparameter setting. In the first table, we varied bucket size B while fixing the number of hash functions to be K=2. In the second table, we fixed the bucket size B to 109, which is the same as in the paper, and changed the number of hash functions. The collision probability increases from right to left for both tables. The results in the first table show the more likely a hash collision, the more unstable the model performance. However, the deterioration is slow, showing the method's robustness to potential hash collisions and various hyperparameter settings. In the second table, although increasing K reduces the probability of hash collisions, increasing K also increases the number of effective parameters (related to model complexity) to fit in the model. It thereby increases the variance of the predictive performance. Thus, we recommend choosing a small K (such as 2-3) that trades off both hash-collision and predictive performance variance. Note when K=1, the hash collision will cause two items to have exactly the same resulting hash embeddings, leading to a high variance among all settings. We will add these results to the ablation section in the revised paper.

| Ablation study on bucket size B | | | |
|---|---|---|---|
| B=40,K=2 | B=60,K=2 | B=80,K=2 | B=109,K=2 |
| 2.83±0.23 | 2.65±0.16 | 2.56±0.10 | 2.58±0.09 |

| Ablation study on the number of hash functions K | | | | |
|---|---|---|---|---|
| B=109,K=1 | B=109,K=2 | B=109,K=3 | B=109,K=4 | B=109,K=5 |
| 2.66±0.34 | 2.58±0.09 | 2.63±0.16 | 2.78±0.18 | 2.76±0.13 |

Table 6: Comparison between updating all categorical columns' embeddings, only updating incremental columns' embeddings, and updating all model parameters. We used collision-free expandable embeddings in the experiments. The first two updating protocols have little difference but updating all parameters sometimes result in performance deterioration, possibly due to catastrophic forgetting in the network weights.

| | Adult (Acc.) | Bank (Acc.) | Mushroom (Acc.) | Covertype (Acc.) | Retail (Err.) |
|---|---|---|---|---|---|
| Update all columns embeddings | 84.7±0.0 | 90.0±0.0 | 98.8±0.0 | 64.1±0.0 | 3.4±0.3 |
| Update incremental columns embeddings (in use) | 84.8±0.0 | 90.1±0.0 | 98.8±0.0 | 64.0±0.4 | 3.2±0.4 |
| Update all model parameters | 83.3±0.1 | 89.5±0.0 | 98.8±0.0 | 64.0±0.1 | 287.9±125.5 |

*Remedy.* We use the standard trick of multiple independent hash functions to reduce the collision probability of two unique items. As is standard in universal hashing [Carter and Wegman, 1997], the probability of collision with all K hash functions each hashing into B buckets is proportional to (see section 3.3). Collision of hash values could happen among popular, important categories. To address this issue, we can select the desired hash functions that avoid important collisions before applying the hash functions. In addition, users come and go fast, and collisions may become unimportant over time.

