# OpenReview forum: "Probabilistic Hash Embeddings for Temporal Tabular Data Streams"
_ICLR.cc/2025/Conference — Submitted to ICLR 2025_

### Official Review · Reviewer_SDqb · 2024-10-29

**Soundness:** 2
**Presentation:** 1
**Contribution:** 3
**Rating:** 5
**Confidence:** 3

**Summary:**

In the paper, the memory efficiency of data stream machine learning is addressed by proposing a hash table for open vocabulary categorical features, which is learned by Bayesian online learning.

The main idea of the paper is to replace a deterministic hash table by a probabilistic one (PHE), in which each cell is a probability distribution and is updated by calculating the posterior distribution. Properties of PHE follow from the theory of online Bayesian learning, notably in each step, the posterior distribution is the same whether learned online or at once, based on all samples so far. In my opinion, this is the key idea, however it is hidden in Appendix D starting at line 1030. Furthermore, no reference is given to the theorem (or hidden and I did not spot).

The idea of using Bayesian online learning in a probabilistic hash table is nice, useful, and up to my knowledge novel.

In practice, I am a bit uncertain about the importance of the hash table size, especially in light of the large language models. More motivation, including data and vocabulary sizes, would be necessary. Data in the experiments are definitely small in this sense.

Unfortunately, the paper is not well written, very hard to follow the presentation jumping back and forth between main paper and Appendix. While the very simple hand picked two-value example is worked out in much detail showing forgetting can occur with partial hash collision (which is not a surprise in my opinion), several important details are not worked out at all. For example, a detailed Bayesian update for the same example would be more enlightening than the deterministic analysis. Specific suggestions are
-    Moving key technical details from the appendix into the main text;
-    Adding a worked example of the Bayesian update to complement the deterministic analysis;
-    Reorganizing the content to reduce the need to jump between main text and appendix.

Most importantly, the Bayesian update step is only hinted for Deep Kalman filters. In equation (2) an independence assumption is used but its validity is not verified. In general, for Bayesian online learning, assumptions on the distribution are necessary, otherwise the posterior calculation for the update requires the knowledge of the entire past data. For the different regression applications, there is not even a sketch on how the update is calculated.  Please add the following technical description:
-    Justify the independence assumption in equation (2).
-    Specify the distributional assumptions required for the Bayesian online learning updates.
-    Provide at least a high-level sketch of how the updates are calculated for the different regression applications.

Minor question: why can we assume the latent time variable z (line 222) is Gaussian?

While the goal of the paper is to close to the upper bound of the collision-free hash and the experimentations show good results in this sense, I am not convinced whether competitive models are used for the tasks. For example, is Deep Kalman filter recommendation competitive to state-of-art recommenders? Why was it selected? Is it because the update is feasible to compute? This is not explained in the paper. Please justify your choice of Deep Kalman filters and other baseline models.

For example, some specific state-of-the-art recommender systems could be included as baselines include ADDVAE, SLIM, or even simple traditional ones as knn, also using some evaluation framework such as https://github.com/enoche/MMRec.

Minor typo: line 978 e(2) and not e(1).

**Strengths:**

- The idea of mitigating hash collision by Bayesian online learning is nice and up to my best knowledge new
- Works well in experiments
- Source code in supplement, expected to be fully reproducible upon acceptance

**Weaknesses:**

- May technical details are missing from the paper, most notably the various versions of the Bayesian online learning step with the underlying assumptions on the distribution
- Paper is difficult to read, reader has to switch back and forth between main text and Appending. A simple example is worked out in much detail on why deterministic hash forgets in case of a partial collision (no big surprise), but there is no similar example that illustrate how the Bayesian update mitigates the effect of a collision.
- The selection of base regressors and recommender methods are not really explained, why for example deep Kalman filters?
- The data used in the experiments are small, for a better motivation some real large data sets would be necessary.

**Questions:**

Does the applicability of PHE depend on the base machine learning method?

How can one compute the Bayesian update of the hash embedding, what assumptions are necessary on the distributions?

Can you list some real data sets and applications where the size of the embedding really matters, especially considering the huge size of the large language models?

---

> ### Author Response · Authors · 2024-11-27
> **(1/2) Official Comment by Authors**
>
> > The main idea of the paper is to replace a deterministic hash table by a probabilistic one (PHE), in which each cell is a probability distribution and is updated by calculating the posterior distribution. Properties of PHE follow from the theory of online Bayesian learning, notably in each step, the posterior distribution is the same whether learned online or at once, based on all samples so far. In my opinion, this is the key idea, however it is hidden in Appendix D starting at line 1030. Furthermore, no reference is given to the theorem (or hidden and I did not spot).
>
> We appreciate the reviewer's constructive suggestions. We mentioned this idea in our introduction section and in the small-scale analytical experiments. We will make this part clearer in our next revision.
>
> > In practice, I am a bit uncertain about the importance of the hash table size, especially in light of the large language models. More motivation, including data and vocabulary sizes, would be necessary. Data in the experiments are definitely small in this sense
>
> A larger vocabulary and embedding table is a strong mark in categorical feature intensive applications like retail, E-commerce, recommendation, healthcare, and medical diagnosis. Although continually expanding the embedding table with rows for new items typically improves accuracy, it comes at a cost. A larger embedding table requires more resources to deploy, reduces memory efficiency, and slows down execution. This is a well documented issue and an active research area [Ko et al., 2022; Shi et al., 2020b; Kang et al., 2021; Coleman et al., 2024].
>
> Hash embeddings provide a condensed embedding representation for sparse features that are either novel or unbounded. The size of the hash embedding is fixed irrespective of the size of the feature vocabulary. The fixed-size nature of hash embeddings facilitates efficient memory allocation and accelerates model execution. This observation is not new and is leveraged by many practical systems [Weinberger et al., 2009; Coleman et al., 2024]. By mapping an extensive and dynamically evolving feature sets into a unified predetermined finite-dimensional embedding space, hash embeddings address the challenges of high-dimensional and sparse feature in machine learning applications. However, previous work only focuses on offline settings.
>
> Therefore, our experimental design focuses on online learning and demonstrates the efficacy of our proposed PHE (Probabilisitic Hash Embeddings) across various real-world applications that have a temporally dynamic vocabulary. Investigations on four science datasets or domains, one retail datasets of a whole-year sales data, and one movie recommendation dataset spanning 30 years aim to demonstrate the plug-in and applicability nature of our method in a dynamic environment of changing categorical features. By examining the performance in such environments, we confirm PHE’s robustness and adaptability in data streams and continual learning tasks.
>
> > Unfortunately, the paper is not well written, very hard to follow the presentation jumping back and forth between main paper and Appendix. While the very simple hand picked two-value example is worked out in much detail showing forgetting can occur with partial hash collision (which is not a surprise in my opinion), several important details are not worked out at all. For example, a detailed Bayesian update for the same example would be more enlightening than the deterministic analysis. Specific suggestions are [...]
>
> We thank you for your suggestions. We will reorganize the content of the appendix and the main paper in our next revision.
>
> > While the goal of the paper is to close to the upper bound of the collision-free hash and the experimentations show good results in this sense, I am not convinced whether competitive models are used for the tasks. For example, is Deep Kalman filter recommendation competitive to state-of-art recommenders? Why was it selected? Is it because the update is feasible to compute? This is not explained in the paper. Please justify your choice of Deep Kalman filters and other baseline models.
>
> It is important to note that our focus is not on improving the SoTA for specific models or tasks, such as sequence modeling or recommendation systems. Rather, our primary objective is to showcase the broad applicability of PHE to a spectrum of models - both deterministic and probabilistic ones. To this end, we have selected the Deep Kalman Filters (DKF) as a representative probabilistic model for sequence modeling. Our empirical study illustrates the versatility of PHE across diverse modeling approaches and data types. We will emphasize this point more clearly in our revision of the draft.

---

> ### Author Response · Authors · 2024-11-27
> **(2/2) Official Comment by Authors**
>
> > Does the applicability of PHE depend on the base machine learning method?
>
> No, it doesn't depend on the base machine learning methods. As demonstrated in our experiments, PHE is applicable to a wide range of applications with various base models. We will emphasize this more clearly in the revised version of our draft.
>
> > How can one compute the Bayesian update of the hash embedding, what assumptions are necessary on the distributions?
>
> The details of the Bayesian updates are provided in Section 3.3, where we apply variational inference to approximate the true posteriors. We assume that the approximate posterior distributions of the embedding entries and latent variables are independently Gaussian, as shown in Eq. 2 and the following paragraph. Such assumptions are common in deep latent variable models and approximate inferences. And we don’t make assumptions on data distributions. By optimizing the objective function in Eq. 5, we learn these approximate posteriors.

---

> > ### Comment · Reviewer_SDqb · 2024-12-01
> > **Answer to authors' official comments 1 and 2**
> >
> > Dear Authors,
> > Thank you very much for your answers to my questions.
> > I wonder when you plan to submit the next revision, I would like to see the changes in the paper.
> > While I understand that the proposed method is independent of the machine learning algorithm, it would be more convincing to experiment with models and larger data, similar to as in [Coleman et al., 2024], which could also experimentally justify the independence and distribution assumptions needed.

---

> > > ### Author Response · Authors · 2024-12-02
> > > **Response to follow-up concerns**
> > >
> > > Thank you for your response. We appreciate the help and suggestions for improving our paper. We will update the paper as soon as possible, but right now we have passed the updating deadline. Rest assured, we will implement the modifications reflected in the paper during the next available window.
> > >
> > > It is important to note that our goal differs from Coleman et al., 2024, and thus we illustrate our method from a different dimension. While Coleman et al., 2024 showcased that their Feature Multiplexing framework leads to improvements in several base embedding representations in an offline setting, our experiments, as the first work in an online learning environment, focus on demonstrating the applicability of our method in various domains and applications. While Coleman et al., 2024 focused on a single recommendation task, our study examined three distinct machine learning tasks utilizing different ML models: classification (multilayer perceptrons), sequence modeling (deep Kalman filters), and recommendation (neural collaborative filtering). Furthermore, our most up-to-date MovieLens-32M dataset (released last year), which spans approximately 30 years and contains 32 million ratings, is significantly more extensive than the MovieLens-1M dataset used by Coleman et al., 2024, which covers roughly 13 years and includes 1 million ratings.
> > >
> > > We believe our empirical investigations on four scientific datasets or domains, one retail dataset of a whole year's sales data, and one movie recommendation dataset spanning 30 years demonstrate the validity of our assumptions in dynamic environments with changing categorical features. Per the reviewer's suggestions, we will also add baselines used in Coleman et al., 2024 to show their performance in online learning environments.

---

### Official Review · Reviewer_sk3c · 2024-11-03

**Soundness:** 2
**Presentation:** 2
**Contribution:** 2
**Rating:** 5
**Confidence:** 3

**Summary:**

This paper proposes a probabilistic hash embedding (PHE) model for handling temporal tabular data-streams (TTD) with both categorical and numerical features, addressing the challenges of unbounded growth and evolving categorical items. The PHE model adapts to new items without forgetting old ones, maintains a fixed parameter size, and efficiently supports incremental learning in both offline and online settings. Experimental results show that PHE outperforms baseline methods in classification, sequence modeling, and recommendation tasks.

**Strengths:**

The research ideas are clear.
The experiments are extensive.

**Weaknesses:**

1. The transitions between paragraphs are quite abrupt, with a lack of logical connections, making some parts difficult to understand. Could the logical structure be further refined?
2.In the “Related Work” section, is the relevance of temporal and recommendation models to the research topic truly that strong? Additionally, could the practical shortcomings of these models be discussed in more detail?
3.There are some minor writing errors. For instance, the formula in line 232 should be numbered.
4.In the “Experiment Overview” section, could you elaborate on the detailed setup of the experiments? Additionally, it is worth noting why the five curves in Figure 5 (Right) exhibit very similar trends.
5.In Table 2, could there be more comprehensive comparisons regarding memory usage?

**Questions:**

1. The transitions between paragraphs are quite abrupt, with a lack of logical connections, making some parts difficult to understand. Could the logical structure be further refined?
2.In the “Related Work” section, is the relevance of temporal and recommendation models to the research topic truly that strong? Additionally, could the practical shortcomings of these models be discussed in more detail?
3.There are some minor writing errors. For instance, the formula in line 232 should be numbered.
4.In the “Experiment Overview” section, could you elaborate on the detailed setup of the experiments? Additionally, it is worth noting why the five curves in Figure 5 (Right) exhibit very similar trends.
5.In Table 2, could there be more comprehensive comparisons regarding memory usage?

---

> ### Author Response · Authors · 2024-11-27
>
> > The transitions between paragraphs are quite abrupt, with a lack of logical connections, making some parts difficult to understand. Could the logical structure be further refined?
>
> Thank you for your suggestions. We will make the presentation better in the next revision.
>
> > In the “Related Work” section, is the relevance of temporal and recommendation models to the research topic truly that strong? Additionally, could the practical shortcomings of these models be discussed in more detail?
>
> We discuss temporal and recommendation models because they are important motivations and applications of condensed embedding representations [Krishnan et al., 2015; Ko et al., 2022; Shi et al., 2020b; Kang et al., 2021; Coleman et al., 2024]. We will rephrase this section to make them more relevant.
>
> > In the “Experiment Overview” section, could you elaborate on the detailed setup of the experiments? Additionally, it is worth noting why the five curves in Figure 5 (Right) exhibit very similar trends.
>
> Thank you for your suggestions. We conducted our experiments in data streaming environments where data points arrive sequentially. Our goal was to simulate practical settings in which new feature items can occur over time and need to be incorporated into the system.
>
> The metric curves for MovieLens-32M in Fig. 5 show similar trends because the experimental environments are consistent across all methods. However, the ranking among the baselines changes over time: AdaFast, despite performing best initially, gradually experiences a decline in performance as more data is revealed.
>
> > In Table 2, could there be more comprehensive comparisons regarding memory usage?
>
> We hope Table 2 provides readers with a concrete understanding of PHE's memory efficiency. The formal memory complexity, which is independent of the vocabulary size, is presented in line 202.

---

> > ### Comment · Reviewer_sk3c · 2024-12-02
> >
> > Thank the authors for the response. This paper is not good enough for ICLR. I maintain my score.

---

> > > ### Author Response · Authors · 2024-12-02
> > >
> > > We thank the reviewer for their time in reviewing and providing suggestions for our paper. We will incorporate these suggestions into the new revision. If you have any further questions or concerns, please don't hesitate to ask us.

---

### Official Review · Reviewer_xeuB · 2024-11-03

**Soundness:** 2
**Presentation:** 1
**Contribution:** 2
**Rating:** 3
**Confidence:** 3

**Summary:**

The paper discusses the problem of learning with temporal tabular data streams. The challenge in this setting is that the vocabulary of categorical data, and the relevance or meaning of categorical data fields can change over time. A fixed embedding of those categorical features, which is typical in offline methods, does not address those challenges in an online setting.

The paper proposed a new bayesian method, called probabilistic hash embedding (PHE), that is able to incrementally learn those embeddings based on the input data observed so far. The final goal is to learn an ML model in the TTD stream setting, exploiting the PHE.

The paper runs experiments to show the advantage of their method with respect to other different ways to update the categorical embeddings.

**Strengths:**

The work addresses an important problem. Tabular data is the most common source of data, and in an online setting, the changes in the vocabulary of categorical features and their meaning are an important problem.

The paper provides an extension to existing prior methods that use a fixed unified embedding for the categorical features, which are not updated over time. I believe that the method proposed by the authors to update those embeddings, based on the input data, is original and significant.

**Weaknesses:**

I found Section 3 unclear and very hard to follow. The mathematical notation is often imprecise, and it is hard to understand what the paper is trying to accomplish. This is one of the paper's main weaknesses, making it hard to appreciate its contribution, and to understand the algorithm presented in the paper. This is the main reason for my final score.

I will give several examples:

In line 182, the hash function is defined to be a number.

E is a $B \times d$ matrix. What does it mean for a matrix E to be Gaussian with a diagonal covariance(see footnote 3)?

What is the approximation in Equation (1)? Why does the expectation become probability?

The discussion 209-215 seems to be out of place. It is not possible to follow comments like "only a few embeddings need to be updated online" since the update is not discussed in Section 3.2. In particular,  Lines 209-215 for the paragraph "Discussion" seem completely unrelated to Section 3.2

Section 3.3 discusses how PHE can be used in conjunction with Deep Kalman Filter. However, Deep Kalman Filter is also never presented or discussed.

In line 224, the function $f_{\theta_z}$ is defined as a set of a vector and a matrix.  A normal distribution $\mathcal{N}$ has two parameters $\mathcal{N}(\mu, \Sigma)$. In lines 223, and 227, what does it mean $\mathcal{N}(z \mid 0,I)$?

In lines 234-235, it is written that the "Observation of other tasks is generated similarly beside the hash-embedding [...]$. This is the first time that the word "task" appears in the paper. What is a task?

In Lines 245: also assume that $q_{\theta}()$ is a Gaussian distribution implemented as a recurrent neural network. What does this mean?

How can the covariance $\Sigma_{\lambda}$ have the same dimensionality $B \times d$ than the vector $\mu_{\lambda}$? How is the covariance $\mu_{\lambda}$ diagonal as written in Lines 243.

Why is the covariance $\Sigma_{i,\phi}$ a vector in line 247.

I would add a reference for Kalman Filters in line 228.

In line 254, is maximizing the ELBO equivalent to maximizing the marginal likelihood? Also, there are no references for these known techniques as ELBO, structured variational inference.

Section 3.4, which is supposed to be theory, only contains a simple example, using simplifying assumptions. Although the paper contains a lot of equations and it is mathematical dense, it has no formal proposition/lemma/thm. It is unclear what Section 3.4 is trying to achieve. I believe theoretical results should be summarized in a formal statement (as a theorem).

----

For the experimental section, there is no comparison with the baseline from previous work. Does this mean that there exists no method in the literature that can be applied in your experimental setting (even if it has low performance?), even ones that do not use unified embeddings?

Since this is an important problem, and in the real world, there are indeed applications where categorical data is used without prior knowledge of the vocabulary size (e.g., advertisement), it seems peculiar that no comparison with previous work appears in the paper. A comparison with previous work would also strengthen the paper since it would further motivate the importance of solving this problem in the context of the existing literature.

---

Usually, hash functions are chosen using a random seed. Why are the methods from prior work that use hashing tricks deterministic, as written in Lines 117-119?

**Questions:**

See Weaknesses for Question on Section 3.

In Section 3, I would recommend clearly stating the assumption, maybe summarizing all the random variables and functions used in the paper using a table, with the correct dimensionality.

---

> ### Author Response · Authors · 2024-11-27
>
> > The mathematical notation is often imprecise, and it is hard to understand what the paper is trying to accomplish.
>
> We thank the reviewer for the suggestions. We will improve the notations in the revised version. Regarding the key points of our paper, we have summarized them in the general comments.
>
> > For the experimental section, there is no comparison with the baseline from previous work. Does this mean that there exists no method in the literature that can be applied in your experimental setting (even if it has low performance?), even ones that do not use unified embeddings? Since this is an important problem, and in the real world, there are indeed applications where categorical data is used without prior knowledge of the vocabulary size (e.g., advertisement), it seems peculiar that no comparison with previous work appears in the paper.
>
> Our work is the first work to handle open vocabulary in online dynamic environments. Appropriate baselines require handling both unbounded feature items and online updates simultaneously. To construct baselines for our setup, we need to combine existing embedding methods designed for offline inference with online updating techniques.
>
> For embedding baselines, we use both Hash or Unified Embeddings (HE) and Expandable Embeddings (EE). While EE is not desirable in practice due to its unbounded memory requirements, we include it as a baseline to understand the performance gap (if any) resulting from our method's memory efficiency. For the online learning baseline, we choose the popular strategy of online fine-tuning by varying the learning rate to give many candidate online learning baselines. The different learning rates correspond to AdaFast/Med/Slow in our paper. All the combinations of the above embedding methods and online learning candidates act as baselines in our experiments.
>
> Our empirical study demonstrates the adaptability of our method in dynamic online learning environments. Specifically, our approach significantly outperforms the hash embedding-based baselines across all applications, while performing on par with the upper-bound baseline.
>
> > Usually, hash functions are chosen using a random seed. Why are the methods from prior work that use hashing tricks deterministic, as written in Lines 117-119?
>
> By “deterministic” we mean their *hash embeddings* are deterministic (vs. probabilistic hash embeddings).

---

> > ### Comment · Reviewer_xeuB · 2024-12-02
> > **Response**
> >
> > I appreciate the authors' reply. I believe that with additional clarity in the presentation of the method, this work has the potential to make a good contribution.

---

> > > ### Author Response · Authors · 2024-12-02
> > >
> > > We thank the reviewer for their time in reviewing and providing suggestions for our paper. We will incorporate the suggestions into the new revision.

---

### Official Review · Reviewer_JXPY · 2024-11-05

**Soundness:** 4
**Presentation:** 3
**Contribution:** 4
**Rating:** 8
**Confidence:** 3

**Summary:**

The paper studies the temporal tabular data streams (TTD), where each observation has both categorical and numerical values and the universe of distinct categorical items is not known upfront and can even grow unboundedly over time. As pointed out by the paper,  feature hashing is commonly used as a preprocessing step in this case but currently such a method is only considered for the offline or batch settings.
In this paper, the authors give a new stochastic method called probabilistic hash embedding (PHE) which applies Bayesian online learning to learn incrementally with data. The paper then shows that PHE is useful in the downstream inference algorithm implementation. Finally, the paper gives an empirical evaluation which demonstrates the advantage of the proposed method.

**Strengths:**

- The technical contribution of this paper is solid. The paper gives a stochastic method for feature embedding, which is based on the Bayesian online learning method while in the previous works that is only considered in the offline or batch settings. This is interesting to me. In addition, the paper also gives a theoretical explanation that shows the advantage of the stochastic embedding method compared to the deterministic embedding method.

- The paper gives a detailed experiment comparison, which demonstrates the advantage of the proposed method.


- The writing of the paper is good and the presentation is clear.

**Weaknesses:**

I am not an expert in the related field, hence currently I do not see some important weakness in this paper.

**Questions:**

N/A

---

> ### Author Response · Authors · 2024-11-27
>
> We sincerely thank the reviewer for evaluating and supporting our work. If you have any further questions, please don't hesitate to ask.

---

### Author Response · Authors · 2024-11-27
**Relevance and importance of our setup in practice**

We sincerely appreciate the time and effort all reviewers have dedicated to evaluating our paper. Before addressing each reviewer's comments individually, we would like to provide the following general responses:

In practice, the presence of a large number of categorical values is a well known issue in machine learning applications. A common and well-established solution to handling them is the use of hashing techniques [Weinberger et al., 2009; Tito Svenstrup et al., 2017; Shi et al., 2020b; Kang et al., 2021; Coleman et al., 2024]. These methods handle high-dimensional categorical data by mapping them to a unified, predetermined lower-dimensional embedding space. However, almost all prior works on hash embeddings focused on offline evaluation where data distributions are stationary and the entire test set was available in a batch fashion.

Our paper considers a more realistic setting in which data arrives in a streaming fashion, without clear train / test split. This setup more closely resembles practical applications where new feature items are continuously observed and must be incorporated into the model in real-time. Additionally, the meaning or behavior associated with existing items may evolve, necessitating an approach that can accommodate these changes.

Ours is the first online method to handle unbounded feature items with a fixed-size model in an online setting. At the core of our approach--PHE--is a simple modification to make the embeddings probabilistic and update them using online Bayesian learning. To demonstrate the efficacy of the proposed method, our experimental design focuses on various real-world applications that have temporally dynamic vocabularies. Investigations on four scientific datasets or domains, one retail dataset of a whole year's sales data, and one movie recommendation dataset spanning 30 years aim to demonstrate the plug-in nature and applicability of our method in a dynamic environment of changing categorical features. By examining the performance in such environments, we confirm PHE's robustness and adaptability in handling data streams and continual learning tasks.

---

### Meta-Review · Area_Chair_vdNh · 2024-12-20

**Metareview:**

The paper proposes a probabilistic hashing scheme for tabular data, where the number of features can grow over time. The method was found to be interesting and shows good results. However, there are several concerns regarding the writing and presentation which were raised by the reviewers. I do not recommend acceptance due to this, but taking the suggestions into account should improve the next revision of the paper.

**Additional Comments On Reviewer Discussion:**

The reviewers pointed out several suggestions to improve the writing. They generally agreed that they would like to see a revision which incorporates these changes.

---

### Decision · Program_Chairs · 2025-01-22

Reject